# MicroRNA-138 controls hippocampal interneuron function and short-term memory in mice

**Reetu Daswani[1], Carlotta Gilardi[1], Michael Soutschek[1], Prakruti Nanda[1], Kerstin Weiss[2], Silvia Bicker[1], Roberto Fiore[1], Christoph Dieterich[3], Pierre-Luc Germain[1], Jochen Winterer[1]\*, Gerhard Schratt[1]\***

[1]Lab of Systems Neuroscience, Institute for Neuroscience, Department of Health Science and Technology, Swiss Federal Institute of Technology ETH, Zurich, Switzerland; [2]Institute for Physiological Chemistry, Biochemical-Pharmacological Center Marburg, Philipps-University of Marburg, Marburg, Germany; [3]Section of Bioinformatics and Systems Cardiology, Department of Internal Medicine III and Klaus Tschira Institute for Integrative Computational Cardiology, University of Heidelberg, Heidelberg, Germany

**\*For correspondence:**
jochen.winterer@hest.ethz.ch
(JW);
gerhard.schratt@hest.ethz.ch
(GS)

**Competing interest:** The authors declare that no competing interests exist.

**Abstract** The proper development and function of neuronal circuits rely on a tightly regulated balance between excitatory and inhibitory (E/I) synaptic transmission, and disrupting this balance can cause neurodevelopmental disorders, for example, schizophrenia. MicroRNA-dependent gene regulation in pyramidal neurons is important for excitatory synaptic function and cognition, but its role in inhibitory interneurons is poorly understood. Here, we identify *miR138-5p* as a regulator of short-term memory and inhibitory synaptic transmission in the mouse hippocampus. Sponge-mediated *miR138-5p* inactivation specifically in mouse parvalbumin (PV)-expressing interneurons impairs spatial recognition memory and enhances GABAergic synaptic input onto pyramidal neurons. Cellular and behavioral phenotypes associated with *miR138-5p* inactivation are paralleled by an upregulation of the schizophrenia (SCZ)-associated *Erbb4*, which we validated as a direct *miR138-5p* target gene. Our findings suggest that *miR138-5p* is a critical regulator of PV interneuron function in mice, with implications for cognition and SCZ. More generally, they provide evidence that microRNAs orchestrate neural circuit development by fine-tuning both excitatory and inhibitory synaptic transmission.

## Editor's evaluation

The authors provide evidence for a role of the micro RNA – mi-138 – in synaptic inhibition and behavior in mice. They designed a novel transgenic model in which a miR-138 sponge is conditionally expressed to sequester endogenous miR-138 in a cell-type specific manner. Using this tool they show that sequestration of miR-138 in a major, parvalbumin-expressing inhibitory cell type results in enhanced inhibitory transmission, and deficits in memory.

## Introduction

MicroRNAs (miRNAs) are short noncoding RNAs that act as negative regulators of mRNA translation and stability (*Bartel, 2018*). Over the last decade, a large body of evidence shows that miRNAs control excitatory neuron development, function, and plasticity (*McNeill and Van Vactor, 2012*; *Schratt, 2009*). Specific miRNAs, for example, miR-132, -134, and -138, have been identified which

control dendritic spines, the major sites of excitatory synaptic contact (*Schratt et al., 2006*; *Siegel et al., 2009*). The complete lack of miRNAs from excitatory forebrain neurons in Dicer-deficient mice enhances learning and memory (*Konopka et al., 2010*), and specific miRNAs have been linked to both short-term and long-term memory regulation (*Cheng et al., 2018*; *Gao et al., 2010*; *Walgrave et al., 2021*). Although miRNAs are likewise substantially expressed in inhibitory γ-aminobutyric acid (GABA) ergic interneurons (*He et al., 2012*), relatively little is known about miRNA function in this cell type. The complete lack of miRNAs reduces the number of cortical interneurons (*Tuncdemir et al., 2015*) while the absence of miRNAs in interneurons expressing vasoactive intestinal peptide (VIP) leads to cortical circuit dysfunction (*Qiu et al., 2020*). miR-128 knockout in Drd1a-positive GABAergic medium spiny neurons of the striatum results in increased motor activity and fatal epilepsy (*Tan et al., 2013*).

In the rodent hippocampus, microcircuits of excitatory pyramidal neurons and local inhibitory interneurons provide an extensively studied model in the context of information processing, with specific implications for the control of spatial short-term and long-term memory (*Booker and Vida, 2018*; *Markram et al., 2004*; *Pelkey et al., 2017*). Among the different interneuron classes, fast-spiking parvalbumin (PV)-expressing interneurons play a particularly prominent role in controlling pyramidal neuron output to drive appropriate behavioral responses (*Murray et al., 2011*; *Rico and Marín, 2011*). For example, CA1 PV interneurons are required for spatial working memory, but neither for reference memory nor for memory acquisition in contextual fear conditioning (CFC) (*Fuchs et al., 2007*; *Lovett-Barron et al., 2014*; *Murray et al., 2011*). PV interneuron dysfunction has been implicated in neuropsychiatric disorders, most notably schizophrenia (SCZ), but also epilepsy and autism-spectrum disorders (ASD) (*Del Pino et al., 2018*; *Sohal and Rubenstein, 2019*). However, the role of specific miRNAs in PV interneurons in the context of higher cognitive function is completely elusive.

## Results

We previously identified the brain-enriched *miR138-5p* as an important regulator of excitatory synapse function in hippocampal pyramidal neurons (*Siegel et al., 2009*). To study the role of *miR138-5p* on a behavioral level, we generated mice with a conditional ROSA26 transgene (138-floxed) which allows expression of a *miR138-5p* inactivating sponge transcript harboring a lacZ coding sequence and six imperfect miR138-5p binding sites (6x-miR-138sponge) upon Cre-recombinase expression. Sponge transcripts sequester endogenous miRNA, thereby leading to miRNA inactivation and the derepression of cognate target genes (*Ebert and Sharp, 2010*). 138-floxed mice without Cre transgene served as a control throughout the experiments. Since these control mice completely lack expression of any sponge-related transcript in vivo, we first carefully validated the specificity and efficiency of the 138-sponge transcript in primary rat hippocampal neurons. Therefore, we compared the 138-sponge transcript to an identical control sponge, except that the six imperfect miR-138 binding sites were replaced with a shuffled sequence not supposed to sequester any miRNA expressed in neurons (cf. Materials and methods). This analysis revealed a robust and specific inactivation of *miR138-5p* by 138-sponge in vitro (*Figure 1—figure supplement 1a-c*).

We then activated 138-sponge expression at embryonic stage in vivo by crossing 138-floxed mice to the ubiquitous Cre-driver line CMV-Cre (138-sponge[ub], *Figure 1a*). lacZ staining revealed highly penetrant expression of 6x-miR-138 sponge in the hippocampus of 138-floxed mice upon CMV-Cre expression (*Figure 1—figure supplement 1d*). To test for the degree of *miR138-5p* inhibition in vivo, we made use of a dual fluorescence miR-138 sensor virus which we injected into the hippocampus of 138-sponge[ub] mice. Briefly, the sensor consists of a GFP coding sequence (cds) followed by two perfect miR-138 binding sites and a mCherry cds (*Figure 1—figure supplement 1e*). Thus, the lower the ratio of GFP/mCherry, the higher the cellular miR-138 activity. Quantifying GFP/mCherry ratios over many infected neurons revealed that in neurons with 6x-miR-138-sponge expression, GFP/mCherry ratios were significantly higher compared to controls, indicative of an efficient sequestering of endogenous *miR138-5p* by our sponge construct (*Figure 1b*).

We next assessed the cognitive abilities of 138-sponge[ub] mice using behavioral testing. Locomotion in the home cage was similar between 138-sponge[ub] and control mice, ruling out severe developmental motor impairments as a potential confound (*Figure 1—figure supplement 1f*). In the Y-maze test, no genotype-dependent differences in spontaneous alternations were observed, suggesting that exploratory behavior was not affected by *miR138-5p* inactivation (*Figure 1—figure supplement 1g*). In contrast, 138-sponge[ub] mice displayed a significant impairment in novelty preference

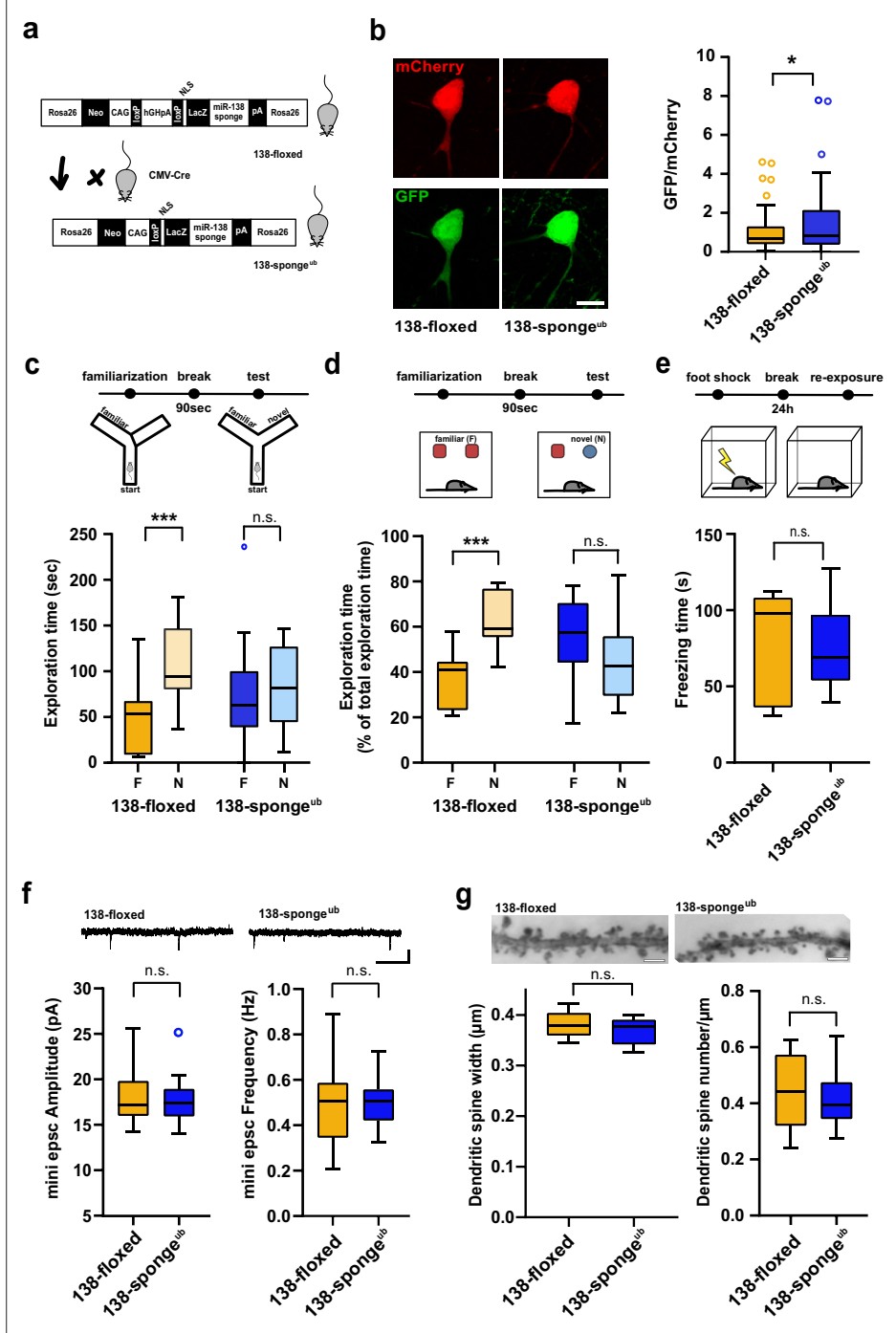

**Figure 1.** Impaired working memory in ubiquitous 138-sponge mice. (**a**) Schematic overview of the strategy for generating 138-sponge[ub] mice. (**b**) Left: representative images of mCherry and GFP expression in hippocampal neurons from 138-floxed and 138-sponge[ub] mice, respectively. Right: bar graphs of GFP/mCherry ratios from CA1 hippocampal neurons infected with a 138-pbds sensor construct; 138-floxed: n=105 cells from two mice, 138-sponge[ub]: n=127 cells from three mice; p=0.02 (KS-test). (**c**) Upper panel: schematic representation of the Y-maze novelty preference task; lower panel: exploration time spent in familiar (F) and novel (N) arm; 138-floxed: n=12 mice; 138-sponge[ub]: n=14 mice; ***p=0.005; n.s.=0.711 (Student's two-tailed heteroscedastic t-test). (**d**) Upper panel: schematic representation of the novel object recognition task; lower panel: exploration time presented as percentage of total time spent with either novel or familiar object; 138-floxed: n=12 mice; 138-sponge[ub]: n=12 mice; ****p<0.00002; n.s. p=0.11 (Student's two-tailed heteroscedastic t-test). (**e**) Upper: schematic representation of the contextual fear conditioning task; lower: time (s) mice spent freezing 24 hr after

*Figure 1 continued on next page*

*Figure 1 continued*

the foot shock was administrated; 138-floxed: n=7 mice; 138-sponge$^{ub}$: n=7 mice; n.s. p=0.97 (Student's two-tailed heteroscedastic t-test). (**f**) mEPSC recording in CA1 pyramidal neurons. Upper panel: example traces; scale bar: 20 pA, 500 ms. Lower panel left: mEPSC amplitude (138-floxed: range, from 14.3 to 25.6 pA; median, 17.2 pA; interquartile range [IQR], 3.9 pA. 138-sponge$^{ub}$: range, from 14.1 to 25.2 pA; median, 17.4 pA; IQR, 3.1 pA; n.s. p=0.74 Student's two-tailed heteroscedastic t-test). Lower panel right: mEPSC frequency (138-floxed: range, from 0.2 to 0.9 Hz; median, 0.5 Hz; IQR, 0.2 Hz. 138-sponge$^{ub}$: range, from 0.3 to 0.7 Hz; median, 0.5 Hz; IQR, 0.1 Hz; n.s. p=0.91 Student's two-tailed heteroscedastic t-test). 138-floxed: n=13 cells/4 mice; 138-sponge$^{ub}$: n=13 cells/3 mice. (**g**) Upper panel: representative images of Golgi-stained CA1 pyramidal neuron dendritic segments of the indicated genotypes. Lower panel: quantification of dendritic spine width (left) and density (number/μm; right) based on Golgi staining; 138-floxed: n=15 cells/3 mice (1312 spines total); 138-sponge$^{ub}$: n=18 cells/3 mice (1687 spines total) (n.s., p=0.25 (width); p=0.49 (density); Student's two-tailed heteroscedastic t-test). mEPSC, miniature excitatory postsynaptic current.

The online version of this article includes the following source data and figure supplement(s) for figure 1:

**Source data 1.** This file contains the raw data on which the graphs in *Figure 1* are based.

**Figure supplement 1.** Validation, behavioural and electrophysiological characterization of ubiquitous miR-138 sponge mice.

**Figure supplement 1—source data 1.** This file contains the raw data on which the graphs in *Figure 1—figure supplement 1* are based.

(*Figure 1c*), indicating a loss of spatial short-term memory. In the novel object recognition (NOR) task, 138-sponge$^{ub}$ mice were unable to discriminate the novel from the familiar object (*Figure 1d*), thereby corroborating the observed short-term memory deficit. In contrast, associative long-term memory, as assessed by classical fear conditioning (*Figure 1e*), as well as anxiety-related behavior (open field, elevated plus maze [EPM]), was not affected by *miR138-5p* inhibition (*Figure 1—figure supplement 1h-j*). Thus, ubiquitous *miR138-5p* inhibition leads to a short-term memory deficit.

We went on to test whether short-term memory impairments in 138-sponge$^{ub}$ mice were associated with alterations in synaptic transmission in hippocampal area CA1 and recorded miniature excitatory postsynaptic currents (mEPSCs) in CA1 pyramidal neurons. Amplitude and frequency of mEPSCs were indistinguishable between 138-sponge$^{ub}$ and control slices (*Figure 1f*, *Figure 1—figure supplement 1k, i*). Likewise, we did not detect any significant alterations in dendritic spine morphology in these neurons using Golgi staining (*Figure 1g*). Finally, we did not observe differences in paired-pulse ratio (PPR), which negatively correlates with presynaptic release probability (*Figure 1—figure supplement 1m*), suggesting that excitatory synaptic transmission at the Schaffer collateral CA1 pyramidal cell synapse was not affected by *miR138-5p* inhibition.

To identify *miR138-5p* target mRNAs and to obtain further insight into the biological function of *miR138-5p* regulated genes, we performed polyA-RNA sequencing with total RNA isolated from hippocampal tissue. Differential gene expression analysis recovered a total of 338 differentially expressed genes (DEGs; 265 upregulated, 73 downregulated) (FDR <0.05; *Figure 2a*). The presence of *miR138-5p* binding sites correlated with increased transcript levels compared with 138-floxed mice (*Figure 2b*). *miR138-5p* 7-mer 1a, and to a lesser extent 7mer-m8 and 8mer sites, predicted significant derepression (*Figure 2—figure supplement 1a*). In total, 56 (21%) of the upregulated, but only 6 (8%) of the downregulated genes harbor *miR138-5p* binding sites within their 3′UTR. Taken together, this data demonstrates that many of the observed upregulated genes are a direct consequence of *miR138-5p* inhibition and further confirms the specificity of the *miR138-5p* sponge transgene.

Gene ontology (GO) term analysis on DEGs from the RNA-seq analysis (*Figure 2c*; *Figure 2—figure supplement 1b*) revealed that many GO terms associated with synaptic function are strongly overrepresented in genes upregulated in the hippocampus of 138-sponge$^{ub}$ mice. In order to specify the origin for the observed gene expression changes, we compared DEGs to single-cell RNA-seq data from different cell types present in the hippocampus (*Zeisel et al., 2018*; *Figure 2d*, *Figure 2—figure supplement 1c*). Surprisingly, we found that those genes which were significantly upregulated in 138-sponge$^{ub}$ mice are strongly enriched in inhibitory GABAergic interneurons. This finding is in line with the observation that excitatory synaptic transmission in hippocampal CA1 was not affected by *miR138-5p* inactivation (*Figure 1f and g*, *Figure 2—figure supplement 1j-l*). Accordingly, many of the upregulated *miR138-5p* targets showed a strong expression signal in different classes of inhibitory

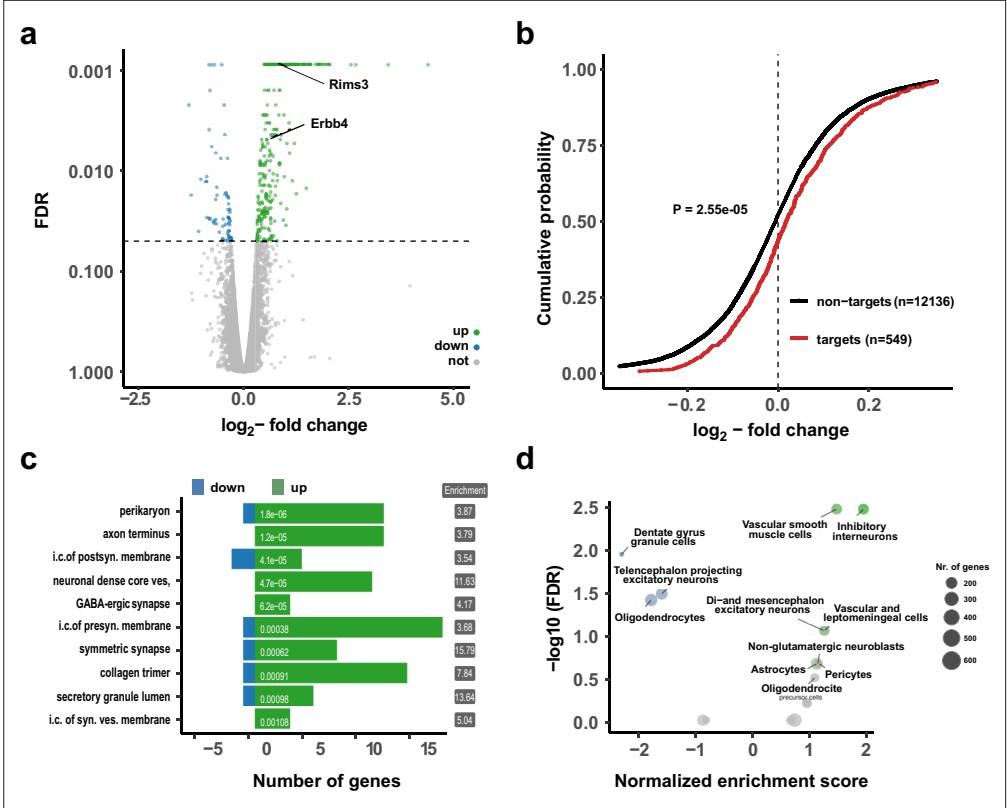

**Figure 2.** Upregulation of interneuron-enriched synaptic genes in ubiquitous miR-138 sponge mice. (**a**) Volcano plot of differentially expressed genes (DEGs) obtained from polyA-RNAseq of total hippocampal RNA from 138-flox and 138-sponge[ub] mice. N=3. Genes with FDR <0.05 are labeled blue (downregulated) or green (upregulated). Rims3 and Erbb4 are indicated. (**b**) Cumulative distribution plots of log$_2$-fold expression changes (138-sponge[ub]/138-floxed) for genes either containing (targets, red curve) or not containing (non-targets, black curve) predicted miR-138 binding sites. p=2.55e$^{-05}$ (KS-test). (**c**) Gene ontology (GO) term analysis for DEGs. Top ten enriched cellular component (CC) GO terms with less than 200 total genes are shown. (**d**) Enrichment analysis of DEGs in different brain cell types based on published single-cell RNA-seq data (*Zeisel et al., 2018*). Normalized enrichment score >0: upregulated in 138-sponge[ub] mice. KS, Kolmogorov-Smirnov.

The online version of this article includes the following source data and figure supplement(s) for figure 2:

**Figure supplement 1.** Gene expression analysis in ubiquitous miR-138 sponge mice.

**Figure supplement 1—source data 1.** This file contains the raw data on which the graphs in *Figure 2—figure supplement 1* are based.

interneurons (*Figure 2—figure supplement 1d*). In contrast, known validated targets of *miR138-5p* which are not enriched in inhibitory interneurons (e.g., Lypla1, Sirt1, and Reln) were not differentially expressed between 138-sponge[ub] and control mice (*Figure 2—figure supplement 1e*).

Next, we performed single-molecule miRNA FISH in cultured rat hippocampal neurons to visualize *miR138-5p* expression at subcellular resolution. This analysis revealed strong expression of *miR138-5p* in both *Camk2a*-positive excitatory and *Erbb4*-positive inhibitory hippocampal neurons (*Figure 3a*). *Erbb4*-positive cells express Gad65, but not Camk2a, confirming their GABAergic phenotype (*Figure 3a*). Quantification of the FISH signal did not reveal any significant differences in the average cellular *miR138-5p* expression between excitatory and inhibitory neurons (*Figure 3—figure supplement 1a*), in agreement with an analysis of published miRNA-seq data from the mouse cortex (*Figure 3—figure supplement 1b*; *He et al., 2012*). In contrast, *miR138-5p* expression was undetectable in GFAP-positive glial cells (*Figure 3—figure supplement 1c*). Thus, *miR138-5p* is robustly expressed in inhibitory interneurons, consistent with a previously unrecognized function in these cells.

We went on to validate *miR138-5p* dependent regulation of predicted interneuron-enriched target genes, focusing on *Erbb4* (Erb-B2 receptor tyrosine kinase 4). *Erbb4* is an SCZ risk gene that has been

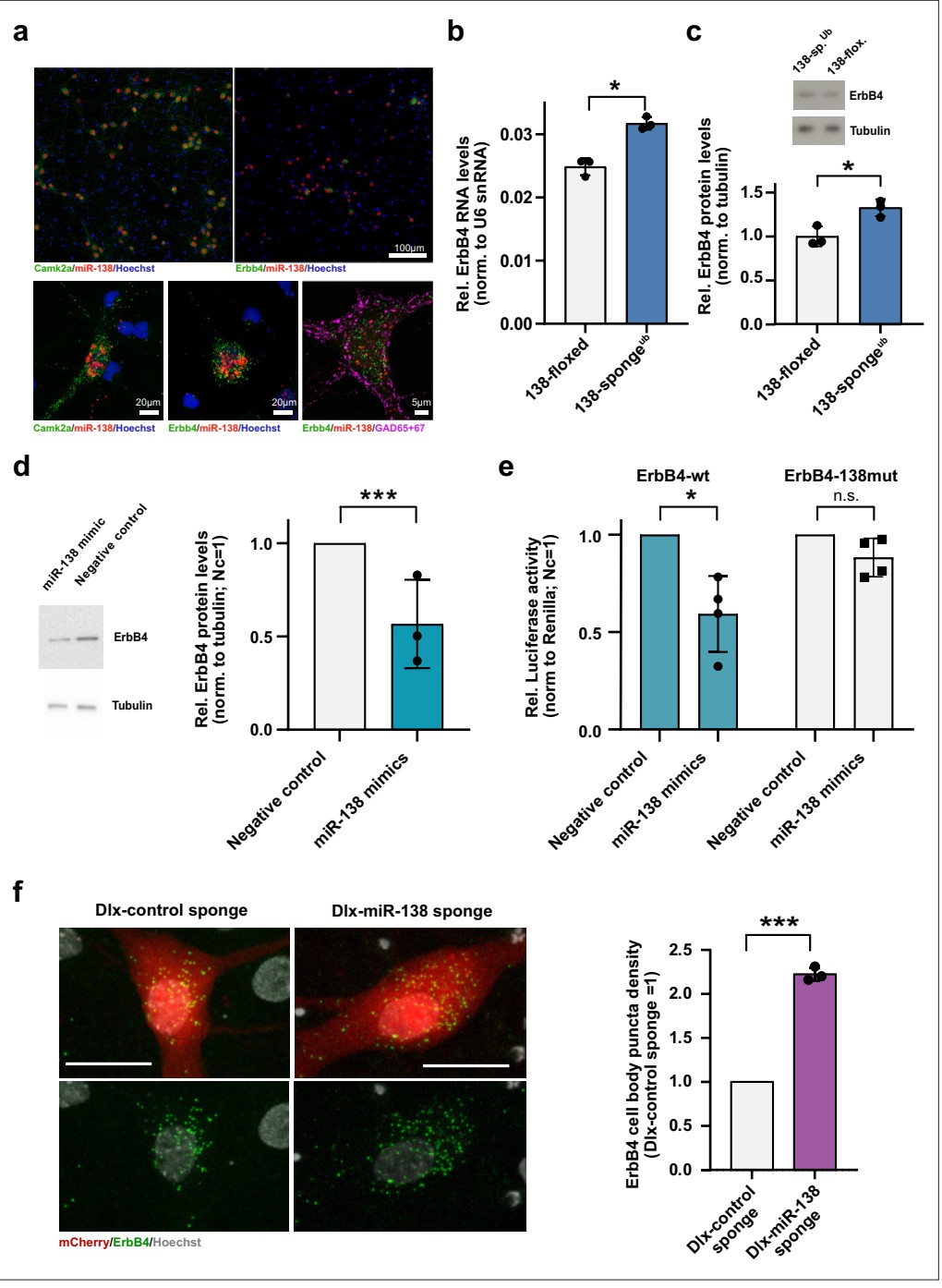

**Figure 3.** Erbb4 is a direct miR-138 target in rat hippocampal interneurons. (**a**) Single-molecule (Sm) FISH analysis of miR-138 (red) together with Camk2a or *Erbb4* mRNA to label excitatory or inhibitory neurons, respectively. Hoechst was used to counterstain nuclei. GAD65/67 antibody staining was used to identify GABAergic neurons. Scale bar =100 μm (upper); 20 μm (lower left and center), 5 μm (lower right). (**b**) qPCR analysis of *ErbB4* mRNA in total hippocampal RNA obtained from 138-floxed or 138-sponge[ub] mice. U6 snRNA was used for normalization. n=3 mice; *p=0.003, (Student's two-tailed heteroscedastic t-test). (**c, d**) Western blot analysis of Erbb4 protein in hippocampal lysates from 138-floxed or 138-sponge[ub] mice (**c**) or lysates from rat hippocampal neurons (DIV12) transfected with miR-138 or control mimic (**d**). Tubulin was used for normalization. (**c**) n=3 mice; *p=0.025 (Student's two-tailed heteroscedastic t-test); (**d**) n=3 independent transfections; ***p=0.0009 (one sample t-test). (**e**) Relative luciferase activity in rat cortical neurons (DIV9-12) transfected with *Erbb4* 3'UTR constructs with (138mut) or without (wt) a mutation in the miR-138 binding site, together with miR-138 or negative control mimics. Negative control

*Figure 3 continued on next page*

*Figure 3 continued*

mimic =1. n=4 independent transfections, *p=0.025, n.s. p=0.09 (Student's two-tailed heteroscedastic t-test). (**f**) Sm FISH analysis of *Erbb4* (green) in rat hippocampal interneurons infected with Dlx-control-sponge or miR-138-sponge. Left panel: representative neurons, scale bar =20 μm. Green: *Erbb4* FISH; gray: DAPI (nuclei); red: mCherry (*Dlx5/6* expressing interneurons). Right panel: *Erbb4* FISH quantification. Control sponge =1. N=3 independent infections (10–12 cells per condition), ***p=0.0004 (one-sample t-test).

The online version of this article includes the following source data and figure supplement(s) for figure 3:

**Source data 1.** This file contains the raw data on which the graphs in *Figure 3* are based.

**Figure supplement 1.** Validation of miR-138 target genes in rat hippocampal interneurons.

**Figure supplement 1—source data 1.** This file contains the raw data on which the graphs in *Figure 3—figure supplement 1* are based.

---

linked to the regulation of GABAergic synaptic transmission and short-term memory (*Fazzari et al., 2010*; *Wang et al., 2018*). Using qPCR and Western blot, we observed a significant upregulation of both *Erbb4* mRNA (*Figure 3b*) and protein (*Figure 3c*) levels in the hippocampus of 138-sponge[ub] compared to 138-floxed mice, thereby validating our results from RNA-seq. On the other hand, transfection of primary rat hippocampal neurons with a synthetic *miR138-5p* mimic reduced Erbb4 protein levels (*Figure 3d*), demonstrating that *miR138-5p* is necessary and sufficient for the inhibition of *Erbb4* expression. In luciferase reporter gene assays, transfection of *miR138-5p* significantly reduced the expression of an *Erbb4* 3′UTR construct containing a wild-type, but not mutant *miR138-5p* binding site (*Figure 3e*), rendering *Erbb4* as direct *miR138-5p* target. Furthermore, we were able to validate the presynaptic vesicle-associated *Rims3* as an interneuron-enriched *miR138-5p* target (*Figure 3—figure supplement 1d-f*).

To study the regulation of *Erbb4* by *miR138-5p* specifically in interneurons, we designed a viral construct in which expression of an inhibitory miR-138 sponge or a respective control sponge is under the control of the interneuron-specific *Dlx5/6* promoter (Dlx-138 sponge and Dlx-control sponge). rAAV expressing Dlx-138 sponge or DLX-control sponge co-localizes with Gad65/67 in primary rat hippocampal neurons, confirming interneuron-specific expression (*Figure 3—figure supplement 1g*). *Erbb4* RNA based on single-molecule fluorescence in situ hybridization (smFISH) was significantly elevated in Dlx-138 sponge compared to Dlx-control sponge infected interneurons (*Figure 3f*), demonstrating that *miR138-5p* cell-autonomously inhibits *Erbb4* expression in hippocampal interneurons.

Our experiments so far suggest that impairments in interneurons are causally involved in short-term memory deficits upon miR-138 inhibition. To address the hypothesis that the observed short-term deficits are hippocampus-dependent, we stereotactically injected Dlx-138 sponge or Dlx-control sponge into the dorsal hippocampus of young adult mice and assessed short-term memory 4 weeks later, as described. Specific expression of 138- and control sponge in hippocampal interneurons was confirmed by immunostaining (*Figure 4a*). The previously observed 138 sponge mediated impairments in Y-maze novel preference and NOR were fully recapitulated by this approach (*Figure 4b and c*), strongly suggesting that *miR138-5p* function in hippocampal interneurons is required for intact short-term memory. Importantly, short-term memory was fully preserved upon hippocampal interneuron-specific expression of the control sponge (*Figure 4b and c*), providing an independent confirmation of the specificity of our approach for *miR138-5p* inactivation.

Next, we investigated electrophysiological alterations that might underlie the observed short-term memory deficits. In hippocampal CA1 pyramidal neurons, which are the main targets of PV-positive (PV+) interneurons in the hippocampal circuit, frequency of miniature inhibitory postsynaptic current (mIPSC) was significantly increased in mice injected with the Dlx-138 sponge as compared to control sponge injected mice (*Figure 4d*). In contrast, amplitude (*Figure 4e*, *Figure 4—figure supplement 1a*) as well as rise and decay time (*Figure 4—figure supplement 1*) of mIPSCs was unaltered.

Based on this observation, we speculated that *miR138-5p* inactivation might be more robust in PV+ interneurons compared to other cell types. Thus, we again injected a miR-138 pbds sensor construct into the hippocampus of control or miR-138 sponge[Ub] mice (*Figure 1b*), but now counterstained for PV to quantify miR-138 activity in PV+ and PV− cells (*Figure 4—figure supplement 1c*). Analysis of GFP/mCherry ratios revealed that expression of a miR-138 sponge significantly inactivated *miR138-5p* in PV+ neurons but had only a marginal effect in PV− cells (which predominantly consist of excitatory pyramidal neurons). Thus, the observed lack of regulation of *miR138-5p* targets in excitatory pyramidal

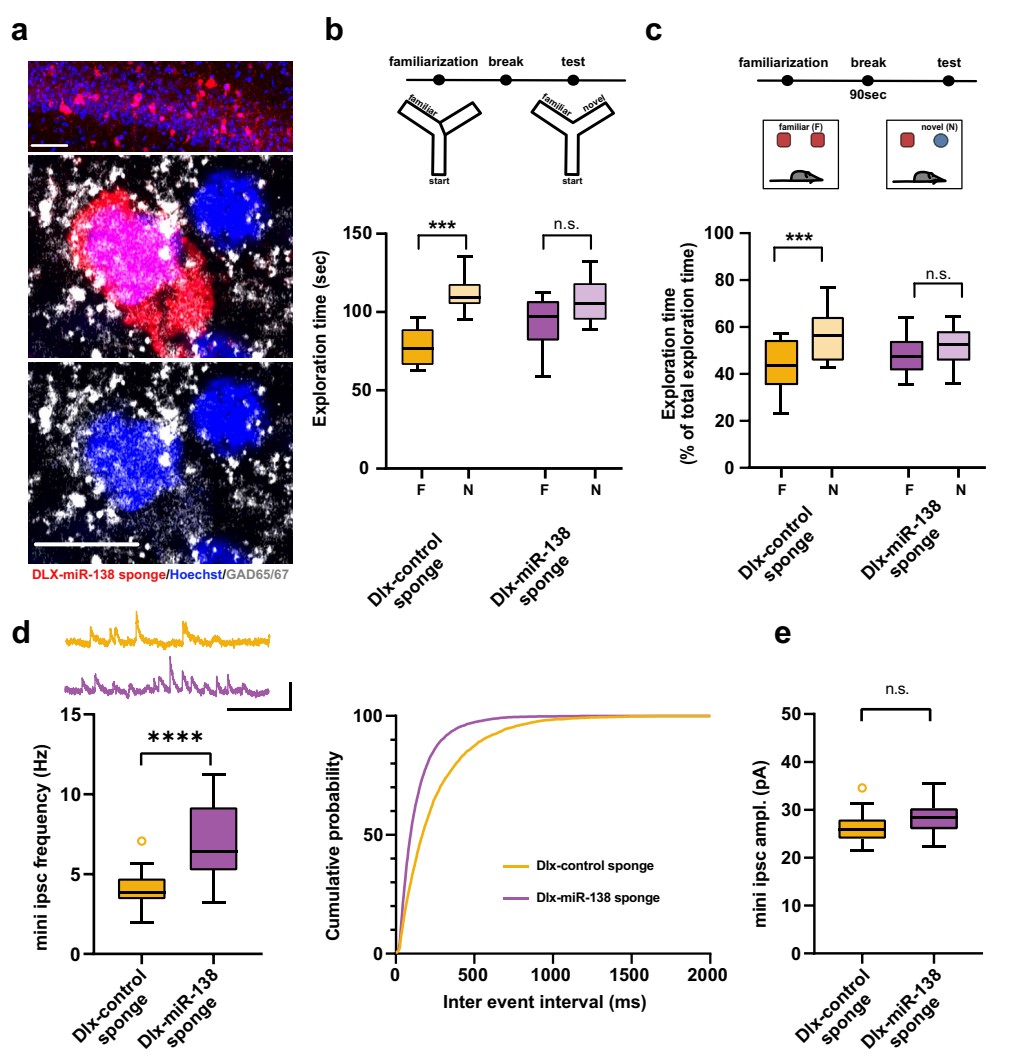

**Figure 4.** Impaired short-term memory and inhibitory synaptic transmission upon miR-138 inhibition in hippocampal interneurons. (**a**) Representative pictures of hippocampal interneurons in vivo infected with rAAV-Dlx-138-sponge. Upper panel: infected interneurons in hippocampal area CA1. Scale bar =50 μm. Middle and lower panels: neurons in hippocampal area CA1 at higher magnification. Left neuron: infected with rAAV-Dlx-138-sponge, expressing GAD65/67; right neuron: not infected, no GAD65/67 expression. Scale bar =10 μm. red: mCherry; gray: GAD65/67, blue: Hoechst nuclei. (**b**) Upper panel: schematic representation of the Y-maze novelty preference task; lower panel: exploration time spent in familiar (F) and novel (N) arm; Dlx-control sponge n=8 mice; Dlx-miR-138 sponge n=9 mice; ****p=0.00005; n.s.=0.079 (Student's two-tailed heteroscedastic t-test). (**c**) Upper panel: schematic representation of the novel object recognition task; lower panel: exploration time presented as percentage of total time spent with either novel or familiar object; Dlx-control sponge n=9 mice; Dlx-miR-138 sponge n=10 mice; *p=0.03; n.s.=0.33 (Student's two-tailed heteroscedastic t-test). (**d**) mIPSC frequency in CA1 pyramidal neurons. Upper panel left: example traces, Dlx-control sponge in orange, Dlx-miR-138 sponge in purple, scale bar: 50 pA, 200 ms. Lower panel left: mIPSC frequency (Dlx-control sponge: range, from 2.0 to 7.1 Hz; median, 3.8 Hz; IQR, 1.3 Hz. Dlx-miR-138 sponge: range, from 3.2 to 11.3 Hz; median, 6.4 Hz; IQR, 4.0 Hz; ****p<0.0001, Student's two-tailed heteroscedastic t-test). Right panel: Cumulative distribution mIPSC frequency (p<0.0001; Kolmogorov-Smirnov test). (**e**) mIPSC amplitude in CA1 pyramidal neurons (Dlx-control sponge: range, from 21.5 to 34.6 pA; median, 25.9 pA; IQR, 4.0 pA. Dlx-miR-138 sponge: range, from 22.3 to 35.5 pA; median, 28.3 pA; IQR, 4.4 pA; n.s. p=0.13 Student's two-tailed heteroscedastic t-test). Dlx-control sponge n=19 cells/2 mice; Dlx-miR-138 sponge n=19 cells/2 mice. IQR, interquartile range; mIPSC, miniature inhibitory postsynaptic current.

The online version of this article includes the following source data and figure supplement(s) for figure 4:

**Source data 1.** This file contains the raw data on which the graphs in *Figure 4* are based.

*Figure 4 continued on next page*

*Figure 4 continued*

**Figure supplement 1.** Inhibitory synaptic transmission upon miR-138 inhibition in hippocampal interneurons and validation of miR-138 sponge activity in PV-expressing hippocampal interneurons.

**Figure supplement 1—source data 1.** This file contains the raw data on which the graphs in *Figure 4—figure supplement 1* are based.

neurons of 138-sponge[Ub] mice is likely a result of inefficient sponge-mediated *miR138-5p* inhibition in this cell type. Since *miR138-5p* and alterations in PV+ interneuron function had recently been linked to SCZ (*Pelkey et al., 2017*; *Watanabe et al., 2014*), we performed a comparison between DEGs from 138-sponge[ub] mouse hippocampus and cortical tissue of SCZ patients (*Gandal et al., 2018*). We found a significant overlap between genes upregulated in the 138-sponge[ub] hippocampus and SCZ patients (p=0.00884; *Figure 4—figure supplement 1d*). Taken together, our results suggest that *miR138-5p* inactivation due to expression of a miR-138 sponge predominantly affects PV+ interneurons leading to an upregulation of synaptic genes which are deregulated in SCZ.

To elaborate on the functional role of *miR138-5p* in PV+ interneurons, we generated 138-sponge[PV] mice by crossing 138-floxed mice to PV-Cre mice (*Figure 5a*). In 138-sponge[PV] mice, the 6x-miR-138-sponge transcript is selectively expressed in the majority (about 90%) of PV-expressing inhibitory interneurons (PV-positive cells) (*Figure 5b*). On a behavioral level, we found that locomotion and anxiety-related behavior, as measured in the open field and EPM tests, was unaltered in 138-sponge[PV] mice compared to their littermate controls (*Figure 5—figure supplement 1a-c*). However, similar as 138-sponge[ub] mice (*Figure 1c and d*) and mice injected with rAAV-Dlx5/6-138-sponge into the hippocampus (*Figure 4b and c*), 138-sponge[PV] mice showed impairments in behavioral tasks addressing short-term memory, such as the Y-maze novelty preference (*Figure 5c*) and NOR (*Figure 5d*) tests. No genotype-dependent differences in spontaneous alternations in the Y-maze were observed (*Figure 5e*). Similar to 138-sponge[ub] mice, associative long-term memory as assessed by conditional fear conditioning was not affected in 138-sponge[PV] mice (*Figure 5—figure supplement 1d*). These results demonstrate that *miR138-5p* activity in PV-positive interneurons is required to sustain proper short-term memory. Next, we investigated electrophysiological alterations that might underlie the observed short-term memory deficits. In hippocampal CA1 pyramidal neurons of 138-sponge[PV] mice, similar to rAAV-Dlx5/6-138-sponge injected mice, frequency of mIPSC was significantly increased in 138-sponge[PV] as compared to control slices (*Figure 6a*, *Figure 6—figure supplement 1a*), while neither amplitude, nor rise or decay time were changed (*Figure 6—figure supplement 1b-d*). The total number of PV-positive interneurons in the hippocampus was similar between 138-sponge[PV] and 138-floxed mice (*Figure 6—figure supplement 1e*), suggesting that mIPSC frequency changes either result from an increased number of inhibitory presynaptic boutons synapsing onto pyramidal cells or an enhanced neurotransmitter release. To distinguish between these possibilities, we first analyzed presynaptic boutons contacting CA1 pyramidal cells by staining slices obtained from 138-floxed and 138-sponge[PV] mice with antibodies against PV and the vesicular GABA transporter (VGAT). Our analysis revealed no significant difference in the number and intensity of PV-positive presynaptic boutons impinging onto the somata of CA1 pyramidal cells between 138-sponge[PV] mice and their littermate controls (*Figure 6b*). To probe for changes in presynaptic release probability, we first recorded extracellularly stimulated inhibitory PPRs (iPPRs) in CA1 pyramidal cells but did not observe differences between the two groups (*Figure 6c*). Finally, we performed paired whole-cell recordings between presynaptic putative fast-spiking PV-positive interneurons in stratum pyramidale and postsynaptic CA1 pyramidal cells (*Figure 6d–f*; *Figure 6—figure supplement 1f*). The analysis of unitary connections revealed increased, albeit not significant, unitary inhibitory postsynaptic current (uIPSC) amplitudes (including failures of transmission) and success rates (i.e., an action potential elicits an IPSC) in 138-sponge[PV] mice as compared to their control littermates (*Figure 6d*). However, we did not observe significant changes in PPR (*Figure 6e*). Our further analysis though revealed a significant decrease of the coefficient of variation (CV), indicative of more reliable unitary synaptic connections between PV-positive interneurons and CA1 pyramidal neurons in 138-sponge[PV] mice (*Figure 6f*). These findings may also explain why mIPSC frequency was found to be altered as these inhibitory currents reflect inputs of many GABAergic neurons onto a single excitatory pyramidal neuron. In conclusion, CA1 pyramidal neurons in 138-sponge[PV] mice receive increased inhibitory GABAergic synaptic input from

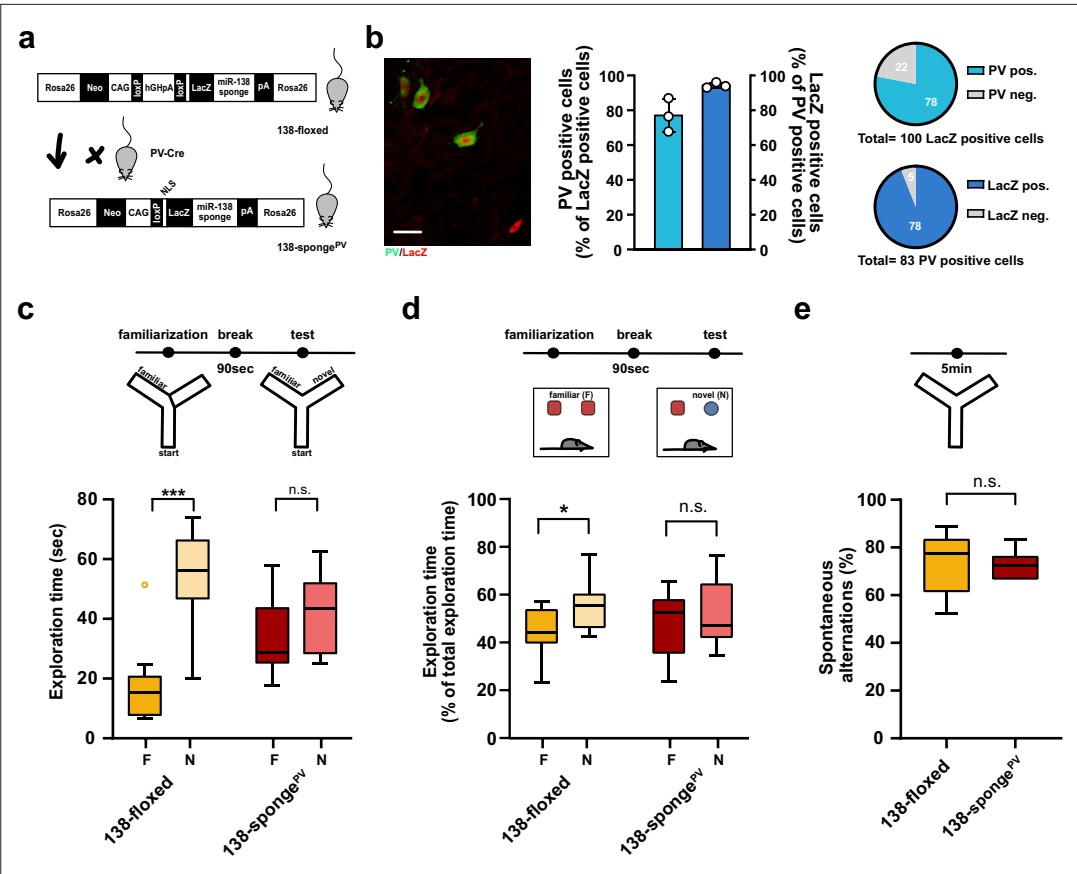

**Figure 5.** Impaired short-term memory in PV-expressing interneuron specific miR-138 sponge mice. (**a**) Schematic overview of the strategy for generating miR-138 sponge[PV] mice. (**b**) Beta-gal expression is largely restricted to PV expressing interneuron. Left panel: representative picture from a CA1 region of a 138-sponge[PV] hippocampal slice co-stained for the lacZ product beta-galactosidase (red) and PV (green). Scale bar =20 μm. Right panel: quantification of PV+/lacZ+ cells in 138-sponge[PV] mice. n=3 mice. (**c**) Behavioral characterization of 138-sponge[PV] mouse line, upper: schematic representation of the Y-maze novelty preference task; lower: exploration time spent in familiar (F) and novel (N) arm; 138-floxed n=10 mice; 138-sponge[PV] n=10 mice; ***p=0.0002; n.s. p=0.19 (Mann-Whitney test). (**d**) Upper: schematic representation of the novel object recognition task; Lower: exploration time presented as percentage of total time spent with either novel or familiar object; 138-floxed n=10 mice; miR-138 sponge n=10 mice; *p=0.035, n.s. p=0.57 (Student's two-tailed heteroscedastic t-test). (**e**) Percentage of spontaneous alternations in the Y-Maze. 138-floxed n=10; 138-sponge[PV] n=10; n.s. p=0.90 (Student's two-tailed heteroscedastic t-test). PV, parvalbumin.

The online version of this article includes the following source data and figure supplement(s) for figure 5:

**Source data 1.** This file contains the raw data on which the graphs in *Figure 5* are based.

**Figure supplement 1.** Behavioural characterization of PV-expressing interneuron specific miR-138 sponge mice.

**Figure supplement 1—source data 1.** This file contains the raw data on which the graphs in *Figure 5—figure supplement 1* are based.

putative fast-spiking PV-positive interneurons without detectable changes in perisomatic inhibitory bouton density or size.

## Discussion

Here, we describe a central role for the brain-enriched miRNA *miR138-5p* in the regulation of inhibitory GABAergic transmission in the hippocampus. Particularly, we find that cell type-specific inhibition of *miR138-5p* in PV-positive interneurons enhances the reliability of unitary synaptic connections from putative fast-spiking PV-positive interneurons onto CA1 pyramidal neurons. The resulting increase in

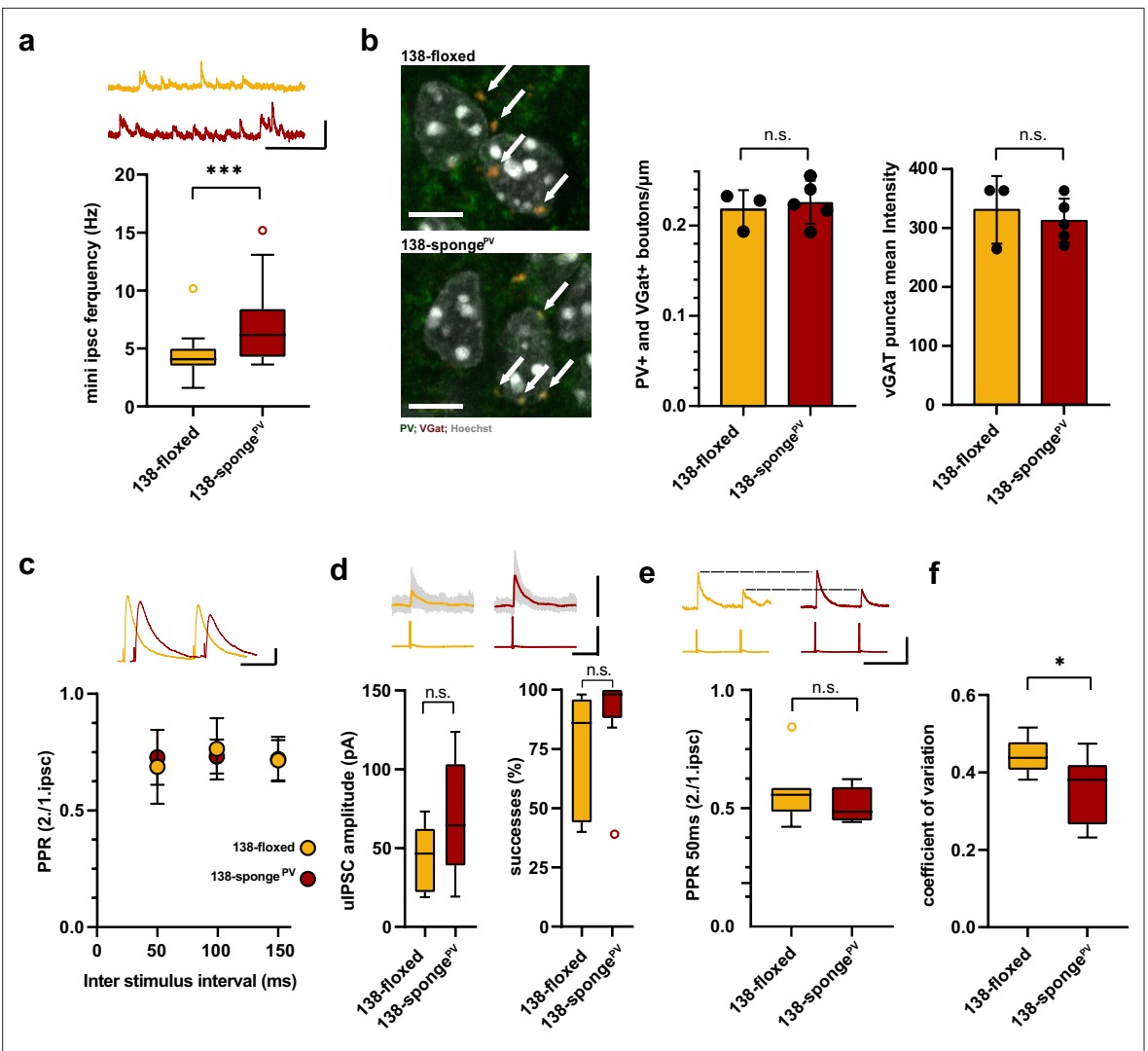

**Figure 6.** Enhanced inhibitory synaptic transmission onto hippcampal pyramidal neurons in PV-expressing interneuron specific miR-138 sponge mice. (**a**) mIPSC frequency in CA1 pyramidal neurons. Upper panel: example traces, 138-floxed in orange, 138-songe[PV] in red, scale bar: 50 pA, 200 ms. Lower panel: mIPSC frequency (138-floxed: range, from 1.6 to 10.2 Hz; median, 4.1 Hz; IQR, 1.5 Hz. 138-sponge[PV]: range, from 3.6 to 15.2 Hz; median, 6.2 Hz; IQR, 4.2 Hz; ***p=0.0002, Mann-Whitney test). 138-floxed n=22 cells/5 mice; 138-sponge[PV] n=23 cells/5 mice. (**b**) PV+, VGAT+ bouton density. Left panel: representative pictures (arrows point the PV+; VGAT+ boutons). Middle panel: number of boutons per CA1 pyramidal neuron cell perimeter based on Hoechst counterstain. Right panel: VGAT puncta mean intensity; 138-floxed: n=73 cells/3 mice; 138-sponge[PV]: n=95 cells/5 mice; data represents the average per mouse±s.d; n.s., p=0.65 (bouton density), n.s., p=0.59 (mean intensity) (Student's two-tailed heteroscedastic t-test); (**c**) Paired pulse ratio (PPR) of stimulated IPSCs in CA1 pyramidal neurons. Upper panel: example traces of PPR (inter stimulus interval of 100 ms) for 138-floxed (orange) and 138-sponge[ub] (red); scale bar: 100 pA, 50 ms. Lower panel: PPRs for different interstimulus intervals ranging from 50 to 150 ms (138-floxed vs. 138-sponge[PV] [mean ±s.d.]: 50 ms, 0.7 ±0.2 vs. 0.7±0.1 [n=13]; 100 ms: 0.8±0.1 vs. 0.7±0.1 [n=13]; 150 ms: 0.7±0.1 [n=12] vs. 0.7±0.1. n.s. p=0.45, p=0.69, and p=0.89 for 50, 100, and 150 ms, respectively. Mann-Whitney test). (**d**) Unitary connections between presynaptic fast-spiking interneurons and postsynaptic CA1 pyramidal cells. Upper panel: example traces, 138-floxed: average of 50 sweeps in orange, 26 single sweeps in gray, 138-songe[PV]: average of 50 sweeps in red, 26 single sweeps in gray; scale bar: 100 pA, 100 mV, 25 ms. Lower panel left: uIPSC amplitude (138-floxed: range, from 18.9 to 73.2 pA; median, 46.6 pA; IQR, 40.2 pA. 138-sponge[PV]: range, from 19.3 to 123.8 pA; median, 64.5 pA; IQR, 64.4 pA; n.s. p=0.11 Student's two-tailed heteroscedastic t-test). Lower panel right: Success rate (138-floxed: range, from 40% to 98%; median, 86%; IQR, 54%. 138-sponge[PV]: range, from 38% to 100%; median, 98%; IQR, 12%; n.s. p=0.14 Mann-Whitney test). 138-floxed n=7 pairs/3 mice; 138-sponge[PV] n=9 pairs/5 mice. (**e**) PPR of unitary connections. Upper panel: example traces, 138-floxed in orange, 138-songe[PV] in red, uIPSCs are normalized to the first uIPSC, scale bar: 100 mV, 50 ms. Lower panel: PPR (2nd/1st uIPSC) (138-floxed: range, from 0.42 to 0.85; median, 0.56; IQR, 0.10. 138-sponge[PV]: range, from 0.44 to 0.62; median, 0.49; IQR, 0.14; n.s. p=0.34 Student's two-tailed heteroscedastic t-test). 138-floxed n=7 pairs/3 mice; 138-sponge[PV] n=9 pairs/5 mice. (**f**) Coefficient of variation (138-floxed: range, from 0.38 to 0.52; median, 0.44; IQR, 0.07. 138-sponge[PV]: range, from 0.23 to 0.47; median, 0.38; IQR, 0.15; *p=0.028, Student's two-tailed heteroscedastic t-test). 138-floxed n=7 pairs/3 mice; 138-sponge[PV] n=9 pairs/5 mice. IQR, interquartile range; mIPSC, miniature inhibitory postsynaptic current; uIPSC, unitary inhibitory postsynaptic current.

*Figure 6 continued on next page*

*Figure 6 continued*

The online version of this article includes the following source data and figure supplement(s) for figure 6:

**Source data 1.** This file contains the raw data on which the graphs in *Figure 6* are based.

**Figure supplement 1.** Electrophysiological characterization of hippocampal neurons in PV-expressing interneuron specific miR-138 sponge mice.

**Figure supplement 1—source data 1.** This file contains the raw data on which the graphs in *Figure 6—figure supplement 1* are based.

neurotransmitter release at this synapse is possibly due to an increase in the number of presynaptic release sites within individual boutons (*Sakamoto et al., 2018*), as we observed a decrease in CV, but did not detect significant changes in release probability. PV-positive interneurons are the main source of feedforward inhibition onto pyramidal neurons (*Pouille and Scanziani, 2001*) and have been functionally linked to working memory (*Murray et al., 2011*), possibly via controlling gamma oscillations (*Hájos et al., 2004*). We therefore propose a model whereby *miR138-5p* activity in PV+ interneurons is regulating neurotransmitter release, thereby keeping pyramidal cell output in a range required for proper information processing. Since *miR138-5p* controls dendritic spine morphogenesis in cultured hippocampal pyramidal neurons (*Siegel et al., 2009*), it might further regulate E-I balance in the hippocampal circuitry by controlling pyramidal neuron excitatory input. The absence of changes in excitatory synaptic transmission in the hippocampus of 138-sponge<sup>ub</sup> mice might be due to ineffective silencing of the highly abundant *miR138-5p* in pyramidal neurons via our approach. This view is supported by our observation that a *miR138-5p* sensor plasmid is effectively unsilenced in PV+, but not PV− hippocampal neurons of 138-sponge<sup>ub</sup> mice (*Figure 4—figure supplement 1d*) and by the lack of upregulation of *miR138-5p* targets preferentially expressed in pyramidal neurons (*Figure 2—figure supplement 1c*). The molecular explanation for this PV-specific *miR138-5p* inactivation is currently unknown.

Sponge-mediated inactivation of miRNAs is routinely used in cell culture, but studies employing this approach in transgenic mice are still scarce (*Giusti et al., 2014*), possibly in part because a stringent control would require the generation of an independent transgenic line. Here, we addressed this issue by comparing the effects of miR-138-sponge to a highly similar, but ineffective control sponge, in primary neurons in vitro and in the hippocampus in vivo using multiple assays. Thereby, we reproducibly observed 138-sponge specific effects on reporter gene expression (*Figure 1—figure supplement 1b*), dendritic spine size (*Figure 1—figure supplement 1c*), miniature inhibitory postsynaptic currents (*Figure 4d and e*), and short-term memory (*Figure 4b and c*), strongly arguing that the effects seen in miR-138-sponge mice are due to *miR138-5p* inactivation.

The molecular mechanisms downstream of *miR138-5p* inactivation leading to enhanced inhibition remain to be elucidated, but likely involve upregulation of proteins organizing inhibitory synapse function. An interesting candidate in this regard represents *Erbb4*, which has been shown to control GABAergic transmission (*Fazzari et al., 2010*; *Wang et al., 2018*) and working memory (*Tian et al., 2017*; *Wen et al., 2010*). In particular, release probability of GABAergic synapses onto pyramidal neurons was diminished in *Erbb4* KO mice (*Wang et al., 2018*), which is in agreement with our finding that elevated Erbb4 levels in miR-138 sponge mice are paralleled by higher mIPSC frequencies. However, additional presynaptic genes might function downstream of *miR138-5p*, for example, *Rims3*, which physically and functionally interacts with presynaptic voltage-dependent Ca²⁺ channels (VDCCs) and increases neurotransmitter release (*Takada et al., 2015*). In the future, restoring the expression of specific inhibitory presynaptic genes in miR-138 sponge mice will be needed to assess their contribution to cellular and behavioral phenotypes caused by miR-138 inhibition.

Intriguingly, interfering with *miR138-5p* in inhibitory neurons of the adult hippocampus fully recapitulates deficits in inhibitory synaptic transmission and short-term memory observed in 138-sponge<sup>Ub</sup> and 138-sponge<sup>PV</sup> mice (*Figure 4*). Although this finding does not rule out a role for *miR138-5p* at early stages of development, it stresses the importance of *miR138-5p* for the homeostasis of the fully developed circuitry. A similar critical role at adulthood in the regulation of GABAergic transmission and behavior was recently shown for the *miR138-5p* target Erbb4 (*Wang et al., 2018*).

Our model of *miR138-5p* being a positive regulator of memory is consistent with results obtained from previous studies performed in mice (*Boscher et al., 2020*; *Tatro et al., 2013*; *Tian et al., 2018*). Moreover, GWAS analysis identified *miR138-5p* as a putative regulator of human memory performance (*Schröder et al., 2014*). A rare miR-138-2 gene variation is furthermore associated with SCZ

in a Japanese population (*Watanabe et al., 2014*), and *miR138-5p* levels are altered in the superior temporal gyrus and dorsolateral prefrontal cortex of SCZ patients (*Beveridge et al., 2010*; *Moreau et al., 2011*). Finally, several *miR138-5p* targets have been genetically linked to SCZ (e.g., *Erbb4*, *Drd2*, and *Igsf9b*) (*Kumar et al., 2010*; *Nicodemus et al., 2006*) or are deregulated in the cortex of SCZ patients (e.g., *Gabra3*, *Fxyd6*, *Tmem132c*, and *Baiap3*, *Gandal et al., 2018*). Thus, *miR138-5p* might be involved in the control of cognitive function in humans and represent a promising target for the treatment of cognitive deficits associated with SCZ and other neurodevelopmental disorders.

# Materials and methods

## Key resources table

| Reagent type (species) or resource | Designation | Source or reference | Identifiers | Additional information |
|---|---|---|---|---|
| Strain, strain background (*Mus musculus*) | miR-138-flox | Taconic Artemis GmbH (Cologne, Germany) | C57BL/6NTac-*Gt(ROSA)26So* $^{tm2459(LacZ, antimir\_138) Arte}$ | |
| Strain, strain background (*M. musculus*) | CMV-Cre | Jackson Laboratories | B6.C-Tg(CMV-Cre)1Cgn (CMV-CRE) | |
| Strain, strain background (*M. musculus*) | PV-Cre | Jackson Laboratories | B6;129P2-*Pvalb*$^{tm1(cre)Arbr}$/J (PV-CRE) | |
| Recombinant DNA reagent | pAAV-6P-SEWB (modified) | *Christensen et al., 2010* | | Used for cloning of miR-138 sponge plasmids |
| Recombinant DNA reagent | pAAV-mDlx-GFP-Fishell-1 | Addgene, 83900 | | Used for cloning of mDlx-miR-138 sponge plasmids |
| Recombinant DNA reagent | pGL3-promoter vector | Promega, Mannheim | | Used for cloning of miR-138 perfect binding site reporter |
| Recombinant DNA reagent | C1-mCherry | Addgene, 632524 | | Used for cloning of miRNA sensor plasmids |
| Recombinant DNA reagent | AAV-hSyn-EGFP | Addgene, 114213 | | Used for cloning of miRNA sensor plasmids |
| Recombinant DNA reagent | pmirGLO dual-luciferase expression vector reporter | Promega, Madison, WI | | Used for cloning of luciferase reporter constructs |
| Sequence-based reagent | Mir-138 FISH probe (Fast Red) | Thermo Fisher Scientific | VM1-10093-VCP | Sequence not provided by commercial supplier |
| Sequence-based reagent | ErbB4 FISH probe (488) | Thermo Fisher Scientific | VC4-3146482-VC | Sequence not provided by commercial supplier |
| Sequence-based reagent | Rims3 FISH probe (488) | Thermo Fisher Scientific | VC4-3146880-VCP | Sequence not provided by commercial supplier |
| Sequence-based reagent | Camk2a FISH probe (488) | Thermo Fisher Scientific | VC6-11639-VCP | Sequence not provided by commercial supplier |
| Sequence-based reagent | Gad2 FISH probe (647) | Thermo Fisher Scientific | VC6-16451-VCP | Sequence not provided by commercial supplier |
| Antibody | aGAD65+67 (rabbit polyclonal) | Abcam | Ab11070 | 1:100 (in vitro); 1:1000 (ex vivo) |
| Antibody | aMAP2 (Mouse monoclonal) | Sigma-Aldrich | M9942 | 1:1000 |
| Antibody | aGFAP (rabbit polyclonal) | Dako | Z0334 | 1:1000 |
| Antibody | aParvalbumin (mouse monoclonal) | SWANT | 235 | 1:1000 |
| Antibody | aVGAT (rabbit polyclonal) | Synaptic systems | 131003 | 1:1000 |
| Antibody | amCherry (rabbit polyclonal) | Abcam | Ab167453 | 1:1000 |
| Antibody | aBeta galatosidase (chicken polyclonal) | Abcam | Ab9361 | 1:4000 |
| Antibody | aMouse Alexa Fluor 488 (Donkey polyclonal) | Invitrogen | A21202 | 1:500 |

*Continued on next page*

*Continued*

| Reagent type (species) or resource | Designation | Source or reference | Identifiers | Additional information |
|---|---|---|---|---|
| Antibody | aRabbit Alexa Fluor 546 (Goat polyclonal) | Invitrogen | A11010 | 1:500 |
| Antibody | aChicken Alexa Fluor 546 (Goat polyclonal) | Invitrogen | A11040 | 1:500 |
| Antibody | aMouse Alexa Fluor 647 (Donkey polyclonal) | Invitrogen | A31571 | 1:500 |
| Chemical compound, drug | Hoechst 33342 | Thermo Fisher Scientific | 62249 | |

## Construct design and cloning

### MiR-138 sponge

MiR-138 sponge was cloned into the BsrGI and HindIII restriction sites of a modified pAAV-6P-SEWB backbone, where the GFP has previously been replaced with dsRed (*Christensen et al., 2010*). Different number of binding sites were tested, and the plasmid containing six binding sites was chosen for further experiments which included the virus production for the creation of the mouse lines.

*MiR-138 sponge imperfect binding site*: 5′ CGGCCTGATTCGTTCACCAGT 3′; spacer sequence: 5′ TTTTT 3′; *Control sponge sequence*: 5′ TGTGACTGGGGGCCAGAGG 3′; spacer sequence: 5′ C AGTG 3′.

### AAV-Dlx5/6-miR-138 sponge

Dlx-138 sponge and Dlx-control sponge were cloned into a modified mCaMKIIα-mCherry-WPRE-hGHp(A) (p199, Viral vector facility of the Neuroscience Center Zurich) backbone where 138-sponge and 138-sponge reversed (control-sponge) were previously inserted using MiR-138 sponge vector as template. The backbone was opened with MluI (NEB: R0198S) and KpnI (NEB: R0142). pAAV-mDlx-GFP-Fishell-1 (gift from Gordon Fishell; Addgene plasmid #83900) was used to cut out Dlx enhancer and HBB promoter with MluI and EheI (NEB: R0606S).

Primers for 138 and control sponge: 5′ CACCACCTGTTCCTGTAGTGTAC 3′; 5′ CCAGAGGTTGAT TATCGATAAGCTTAAC 3′.

### AAV-CMV-Cre

AAV expressing Cre-recombinase under the control of the CMV promoter (gift from Fred Gage; Addgene plasmid #49056).

### pGL3-138 perfect binding site (pbds) vector

The oligonucleotides were designed without any specific overhangs to allow blunt-end cloning in both directions for a perfect binding site reporter and an antisense control reporter. They were annealed and ligated into a pGL3 promoter vector (Promega, Mannheim) expressing firefly luciferase. For cloning, the pGL3 vector was before digested with XbaI and the ends were blunted to insert the binding sites via blunt-end cloning at the 3′end of the firefly open reading frame. After cloning, several clones were sequenced to determine sense and antisense reporter.

### 138 pbds reporter FW

5′CTAGACGGCCTGATTCACAACACCAGCTACCGGCCTGATTCACAACACCAGCTGGATCC 3′.

### 138 pbds reporter REV

5′CTAGCGGATCCAGCTGGTGTTGTGAATCAGGCCGGTAGCTGGTGTTGTGAATCAGGCCGT 3′.

### AAV-miR-138 dual sensor construct

An EcoR1-BglII PCR fragment spanning the pCMV promoter, mCherry coding sequence, and SV40 poly adenylation signal were amplified using C1-mCherry (Addgene, 632524) as template. The PCR product was cloned into the BGlII/EcoR1 sites of the AAV-hSyn-EGFP (Addgene, 114213), upstream

of the human Synapsin 1 promoter. Two miR-138 perfectly complementary binding sites were cloned into the SpeI and HIndIII restriction sites, between the EGFP coding sequence and WPRE element of AAV-hSYN-GFP, to generate the final AAV-138 sensor.

### 138 perfect binding sites FW
5'CTAGACGGCCTGATTCACAACACCAGCTACCGGCCTGATTCACAACACCAGCTGGATCC 3'.

### 138 perfect binding site REV
5'CTAGCGGATCCAGCTGGTGTTGTGAATCAGGCCGGTAGCTGGTGTTGTGAATCAGGCCGT 3'.

### Luciferase constructs
pmirGLO dual-luciferase expression vector reporter (Promega, Madison, WI) was used to clone portions of 3' untranslated regions of the investigated mRNAs. XhoI and SalI restriction enzymes were used.

### ErbB4 wild-type sequence
5'TTGAATGAAGCAATATGGAAGCAACCAGCAGATTAACTAATTTAAATACTTC 3'.

### ErbB4 138-mutant sequence
5'TTGAATGAAGCAATATGGAAGCAgCatGaAGATTAACTAATTTAAATACTTC 3'.

### Rims3 wild-type sequence
5'GCCTCAGTCACCAGCTCTGTACCAGCAATACTCACCCCTCCACCTCCCTGACTT 3'.

### Rims3 138-mutant sequence
5'GCCTCAGTgAtatcCTCTGTACCAGCAATACTCACCCCTCCACCTCCCTGACTT 3'.

## Primary neuronal cell culture
Cultures of dissociated primary cortical and hippocampal neurons from embryonic day 18 (E18) Sprague-Dawley rats (Janvier, France) were prepared and cultured as described previously (*Schratt et al., 2006*). Animal euthanasia was approved by the local cantonal authorities (ZH196/17).

## Transfection
Transfection of primary neurons was performed using Lipofectamine 2000 (Invitrogen, Karlsruhe). For each well of a 24-well plate, a total of 1 µg DNA was mixed with a 1:50 dilution of Lipofectamine in NB/NBP medium. After an incubation of 20 min at room temperature, it was further diluted 1:5 in NB/NBP medium and applied to the cells. Neurons were incubated for 2 hr with the mix. A 1:1000 dilution of APV (20 mM) in NB+/NBP+ was applied for 45–60 min afterward before exchanging with NB+/NBP+.

## Luciferase assay
Luciferase assays were performed using the dual-luciferase reporter assay system on a GloMax R96 Microplate Luminometer (Promega). pmirGLO dual-luciferase expression vector reporter (Promega, Madison, WI) was used to clone portions of 3' untranslated regions of the investigated mRNAs. PcDNA3 was used to balance all amounts to a total of 1 µg DNA per condition.

## Animal lines
The C57BL/6NTac-*Gt(ROSA)26So* tm2459(LacZ, antimir_138) Arte (hereafter named '138-floxed') mouse lines were created at TaconicArtemis GmbH (Cologne, Germany). The targeting strategy allowed the generation of a constitutive LacZ-miRNA138 Sponge Knock-In (KI) allele in the C57Bl/6 mouse ROSA26 locus via targeted transgenesis. The presence of the loxP-flanked transcriptional STOP cassette is expected to terminate the transcription from the CAG promoter and thus prevent the expression of the NLS-LacZ miRNA138 Sponge cDNA, which allows this line to be used as a control (138-floxed line). The constitutive KI allele was obtained after Cre-mediated removal of the loxP-flanked transcriptional STOP cassette from the conditional KI allele, by crossing the 138-flox line with a B6.C-Tg(CMV-Cre)1Cgn

(CMV-CRE) line, which allows the expression of the sponge construct (138-sponge[ub] mice). In all experiments, heterozygous male mice were used. The 138-sponge[PV] line was generated by crossing 138-floxed line with a previously characterized B6;129P2-*Pvalb[tm1(cre)Arbr]*/J (PV-CRE) line.

## Behavioral experiments

Animals were housed in groups of 3–5 per cage, with food and water ad libitum. All experiments were performed on adult male mice (3–5 months old).

The animal house had an inverted light-dark cycle, all testing was done during the dark phase (8 a.m. to 8 p.m.). Mice were handled for 10 min for 5 days before the experiments began. All measures were analyzed by Noldus Ethovision xt 14, unless stated differently. During the experimental phase, mice were transported individually and allowed to acclimatize to the experimental room in a holding cage for at least 20 min before the beginning of the task. At the end of each experiment, they were transported back into the animal storage room in their holding cage and placed back into their original home cage with their littermates. 10 ml/L detergent (For, Dr. Schnell AG) was used to clean equipment in between trials. Tasks which required the use of the same apparatus were scheduled at least 4 days apart. Two separate cohorts of mice were tested ( seven mice each). No cohort-specific differences were found. The behavioral essays were performed from the least to the most stressful: home cage activity, open field, y-maze, EPM, NOR, and CFC. All animal experiments were performed in accordance with the animal protection law of Switzerland and were approved by the local cantonal authorities (ZH017/18).

### Open field (OFT)

OFT was performed as described previously (*Lackinger et al., 2019*). Each session lasted 30 min.

### Y-maze

Spontaneous alternation: mice were placed in a Y-shaped maze (8.5 cm width × 50 cm length × 10 cm height) for 5 min. They were free to explore the whole maze and the alternation between the arms was calculated.

Novelty preference test: mice were given 5 min to explore two arms of the maze during the familiarization phase. A door made from the same Plexiglas used for the walls was used to prevent the access of the subjects into the third arm. At the end of the familiarization phase, mice were placed in their holding cage for 90 s, while the apparatus was cleaned to avoid olfactory trails. Mice were then placed back into the maze for the test phase, where all three arms were accessible. Preference ratio between the familiar and new arm was then scored based on the time spent in those arms.

### Elevated plus maze

The maze was elevated 60 cm from the floor, with two arms enclosed by dark Plexiglas walls (5 cm width×30 cm length×15 cm height), two opposing open arms, and a central platform/intersection. Experiments were conducted in a homogenously illuminated room, with the maze placed in the center of the room. Mice were placed in the central platform of the EPM. The position and motion of the animals were automatically determined and recorded for 5 min. Time spent and distance traveled in the different arms were scored.

### Novel object recognition

Objects were based on the Nature protocol published previously (*Leger et al., 2013*). They were tested beforehand to assess that no object was preferred, and they were randomized between trials and genotypes. Subjects were placed in the open field arena with two items of the same objects for a 5-min familiarization period. After a 90-s break in their holding cage while the arena and the objects were cleaned and one object changed, the mice were put back in the arena for the test phase for 5 min. Time spent exploring the new object was scored manually. Scoring took into consideration the time spent exploring the two objects and the number of visits (nose of the experimental mouse within a 3-cm radius from the object).

### Contextual fear conditioning

This task was performed as previously described (*Siegert et al., 2015*). Briefly, mice were placed in the open field arena with Plexiglas walls and a metal grid bottom inside the multiconditioning

chambers and a metal grid bottom (TSE fear conditioning system, TSE Systems, Germany). They were habituated for 3 min, then foot-shocked (2 s, 0.8 mA constant current) and returned to their home cages. After 24 h, mice were placed in the conditioning chamber. Freezing, defined as a total lack of movement except for heartbeat and respiration, was scored for a 3-min period.

## RNAseq
### RNA extraction
Adult male mice (4 months old) were anesthetized with isoflurane (Baxter, Unterschleißheim, Germany) and then quickly cervically dislocated and decapitated. Hippocampal tissue was dissected and freshly processed. RNA was extracted using mirVana microRNA Isolation Kit (Life Technologies) according to the manufacturer's instructions.

Stranded polyA+ enriched RNA sequencing libraries were prepared at the GENCORE (EMBL, Genomics Core Facility, Heidelberg, Germany) and sequenced on an Illumina HiSeq 2000 machine using a 50-nt paired-end protocol (*Lackinger et al., 2019*).

We removed the sequencing adapter and quality-trimmed all short reads from the 3'end using FLEXBAR 2.5.3 and a Phred score cutoff >10. All reads longer than 18 bp were retained and rRNA reads were subtracted in silico using bowtie2 and an index of the complete repeating rRNA unit (BK000964.3). We used the STAR aligner 2.4.2a4 to map against the mouse genome (EnsEMBL 79 genome+annotations). We performed differential gene expression analysis using Cuffdiff 2.2.1.

## Quantitative real-time PCR
qPCR was performed as described earlier (*Valluy et al., 2015*). Primer sequences are given below.

    ErbB4 FW: 5'GACTCCAATAGGAATCAGTTTGTC 3'.
    ErbB4 REV: 5' TACTGGAGCCTCTGGTATGG 3'.
    Rims3 FW: 5' GCATCAGCGGTGAGATCTGT 3'.
    Rims3 REV: 5' CTGGGTCAAGCCGACGATAG 3'.

## Imaging
Image acquisition was done with the experimenter being blinded to the different conditions. Pictures were taken on a confocal laser scanning microscope equipped with an Airyscan detector (LSM880, Zeiss). Image analysis was carried out on Fiji (ImageJ).

### PV bouton count
A 63× oil objective was used to take images from the CA1 hippocampal region immunostained with PV, VGAT and Hoechst were used to counterstain the nuclei (further details in the immunostaining section). The number of PV+ and VGAT+ en passant boutons around the nuclear perimeter were counted and normalized to the total length of the perimeter. vGAT puncta mean intensity: vGAT puncta mean intensity: ImageJ was used to measure the intensity of all the vGAT puncta in the pictures used to quantify PV boutons.

### PV cell quantification
A 20× objective was used and PV+ cells were counted on a maximum projection intensity of the CA1 region immunostained with PV (Hoechst was used to counterstain nuclei; further details in the immunostaining section). The number of PV + cells was normalized to a defined region of interest (230 × 460 µm$^2$).

### Viral mir-138 sensor
Tilescans were taken with a 20× objective of the infected hippocampal region. Intensity of GFP signal was normalized on the mCherry signal per cell. For the analysis, either all neurons (*Figure 1b*) or only PV+ neurons (*Figure 4—figure supplement 1c*) within the CA1 region were taken into consideration.

## Spine analysis

### In vitro

Hippocampal neurons were transfected at DIV 10 with 200 ng of GFP and the indicated amount of either miR-138-6×-sponge, CTR sponge, AAV backbone, or the Cholesterol-modified 2'O-Me-oligonucleotides ('antagomirs', Thermo Fisher Scientific, Karlsruhe). To balance all amounts to a total of 1 µg DNA per condition, PcDNA3 was used. At 18 DIV, cells were fixed using 4% PFA for 15 min and mounted on glass slides using Aqua-Poly/Mount (Polysciences Inc, Eppelheim). The experimenter was blinded to all the conditions. The z-stack images of GFP-positive neurons exhibiting pyramidal morphology were taken with the 63× objective of a confocal laser scanning microscope (Carl Zeiss, Jena). Eight consecutive optical sections of the dendrites were taken at a 0.4-µm interval with a 1024×1024 pixel resolution. Spine volumes were analyzed with the ImageJ software using the maximum intensity projections of the z-stack images. The GFP intensity of 150–200 spines per cell was measured and normalized to the total intensity of the dendritic tree. For each experimental condition, at least 18 representative neurons derived from three independent experiments were analyzed.

### In vivo

Brains from 3-month-old 138-floxed and 138-sponge[ub] mice were processed with FD Rapid GolgiStain Kit (Gentaur GmbH, PK401) according to the manufacturer's protocol. Pictures were taken with a Zeiss axio-observer seven inverted widefield fluorescence microscope equipped with an Axiocam 702 mono Zeiss camera with a 100× oil objective. Dendritic spines were manually analyzed with Fiji (ImageJ).

## Electrophysiology

Hippocampal slices (300-µm thick) were prepared at 4°C, as previously described (*Winterer et al., 2019*), from 138-floxed, 138-sponge[ub], and 138-sponge[PV] mice (age: 6–8 weeks) and incubated at 34°C in sucrose-containing artificial cerebrospinal fluid (sucrose-ACSF, in mM: 85 NaCl, 75 sucrose, 2.5 KCl, 25 glucose, 1.25 NaH2PO4, 4 MgCl2, 0.5 CaCl2, and 24 NaHCO3) for 0.5 hr, and then held at room temperature until recording.

Whole-cell patch-clamp recordings were performed at 32°C on an upright microscope (Olympus BX51WI) under visual control using infrared differential interference contrast optics. Data were collected with an Axon MultiClamp 700B amplifier and a Digidata 1550B digitizer and analyzed with pClamp 11 software (all from Molecular Devices). Signals were filtered at 2 kHz for mEPSCs and mIPSCs and digitized at 5 kHz. Stimulus evoked and unitary postsynaptic currents were filtered at 4 kHz, fast-spiking interneurons at 10 kHz, both were digitized at 100 kHz. Recording pipettes were pulled from borosilicate capillary glass (Harvard Apparatus; GC150F-10) with a DMZ-Universal-Electrode-Puller (Zeitz) and had resistances between 2 and 3 MΩ.

The extracellular solution (ACSF) was composed of (in mM) 126 NaCl, 2.5 KCl, 26 NaHCO$_3$, 1.25 NaH$_2$PO$_4$, 2 CaCl$_2$, 2 MgCl$_2$, and 10 glucose. For EPSCs measured in CA1 pyramidal cells and for fast-spiking interneurons, the intracellular solution was composed of 125 K-Gluconate, 20 KCl 0.5 EGTA, 10 HEPES, 4 Mg-ATP, 0.3 GTP, and 10 Na$_2$-phosphocreatine (adjusted to pH 7.3 with KOH). Cells were held at –70 mV for mEPSCs and at –60 mV for stimulus evoked EPSCs (eEPSCs). 1 µM Gabazine was added to isolate AMPA-mediated postsynaptic currents, additionally 1 µM TTX for mEPSCs. For IPSCs intracellular solution was composed of 135 Cs-Gluconate, 5 KCl, 2 NaCl, 0.2 EGTA, 10 HEPES, 4 Mg-ATP, 0.3 GTP, and 10 Na$_2$-phosphocreatine (adjusted to pH 7.3 with CsOH). Cells were held at +10 mV for mIPSCs and at –10 mV for stimulus evoked inhibitory postsynaptic currents (eIPSCs) and uIPSCs. 25 µM AP-5, 10 µM NBQX, and 10 µM SCH 50911 were added to isolate GABAa-mediated postsynaptic currents for eIPSCs, additionally 1 µM TTX for mIPSCs. Synaptic currents were evoked by mono-polar stimulation with a patch pipette filled with ACSF and positioned in the middle of CA1 stratum radiatum for eEPSCs and in stratum pyramidale for eIPSCs. Series resistance of CA1 pyramidal neurons (not compensated; range, from 7.0 to 19.7 MΩ; median, 11.3 MΩ; IQR, 2.4 MΩ) was monitored and recordings were discarded if series resistance changed by more than 20%. Membrane potentials were not corrected for liquid junction potential.

For paired recordings, whole-cell configuration was first established in putative fast-spiking interneurons. Cells were selected based on morphological appearance in stratum pyramidale of hippocampal CA1, on fast-spiking properties and on input resistance characteristics for PV-positive interneurons (*Que et al., 2021*; *Figure 6—figure supplement 1f*). Subsequently, whole-cell recordings were made

from postsynaptic CA1 pyramidal neurons residing in close proximity to the presynaptic fast-spiking interneuron. Presynaptic fast-spiking interneurons were held in current-clamp mode and series resistance was compensated with the automatic bridge balance of the amplifier. The mean resting potential of the presynaptic fast-spiking interneurons was –65±4.1 mV (mean ± sd; *Figure 6—figure supplement 1f*). Fast-spiking interneurons were stimulated at 1 Hz for uIPSCs and at 0.03 Hz for paired-pulse uIPSCs. To characterize the discharge behavior of fast-spiking interneurons, depolarizing steps (50 pA) of 1500 ms were applied. The spiking frequency (*Figure 6—figure supplement 1f*) was determined at 800–1000 pA current injection. Input resistance was calculated from the mean deviation from baseline of steady-state voltage responses evoked by –150, –100, and –50 pA current injections.

## Cell type enrichment analysis

For the cell type enrichment analysis, we used the single-cell data and annotation from *Zeisel et al., 2018*. We downloaded single-cell count data and annotation from https://storage.googleapis.com/linnarsson-lab-loom/l5_all.loom, restricted it to hippocampus cells, and aggregated to the pseudo-bulk level using muscat (*Crowell et al., 2020*) and the authors' cell identities. We retained only cell types represented by more than 50 cells, and normalized using TMM (*Robinson and Oshlack, 2010*). For each gene, we then identified the cell type in which it was the most highly expressed, forming gene sets for each cell type. We then created a sponge differential expression signature by multiplying the sign of the foldchange with the –log10(p value) of each gene and looked for enrichment of gene sets in this signature using fgsea (*Korotkevich et al., 2021*).

## GO enrichment analysis

The R Bioconductor package TopGo (v.2.42.0) was used to perform GO enrichment analysis, essentially as in *Lackinger et al., 2019*.

To sum up, genes were annotated with the 'Cellular Component' ontology and significantly changed genes subsequently tested against the expressed background. For the main figure, we plotted the Top10 GO-Terms ranked by significance (filtering out those with more than 200 annotated genes).

## miRNA binding site analysis

Mouse 3'UTR positions of predicted conserved microRNA binding sites were downloaded from Targetscan (version 7.2) (*Agarwal et al., 2015*) and filtered for the seed of *miR138-5p*. Putative miR-138 targets (including site-type information) were then aligned with the genes and log fold changes (logFC) as obtained by the differential expression analysis. Plotted is the cumulative proportion (logFC rank (–1)/number of genes (–1)) over the logFC.

## Single-molecule fluorescence in situ hybridization

smFISH for miRNA detection on hippocampal neuron cultures was performed using the QuantiGene ViewRNA miRNA Cell Assay Kit (Thermo Fisher Scientific) according to the manufacturer's protocol with slight modifications. To preserve dendrite morphology, protease treatment was reduced to a dilution of 1:10,000 in phosphate-buffered saline (PBS) for 45 s. smFISH for mRNA detection was performed using the QuantiGene ViewRNA ISH Cell Assay Kit (Thermo Fisher Scientific) as previously described (*Valluy et al., 2015*), but omitting the protease treatment. After completion of the FISH protocol, cells were washed with PBS, pre-blocked in gelatin detergent buffer, and processed for immunostaining. Pictures represent maximum intensity projections of z-stack images taken on a confocal laser scanning microscope equipped with an Airyscan detector (LSM880, Zeiss).

## Stereotactic surgery

The viral vector (aaAAV-9/2 [hCMV-mCherry-SV40p(A)]rev-hSyn1-EGFP-2x (miR138-5p)-WPRE-SV40p(A)) was produced by the local Viral Vector Facility (VVF9 of the Neuroscience Center Zurich). The produced virus had a physical titer of 6.1×10e12 vector genomes/ml. Stereotactic brain injections were performed on 2- to 3-month-old 138-floxed and 138-sponge[ub] mice as previously described (*Zerbi et al., 2019*). Briefly, mice were anesthetized with isoflurane and subsequently placed onto the stereotaxic frame. Before and after the procedure, animals received subcutaneous injection of 2 mg/kg Meloxicam for analgesia and local anesthetic (Emla cream 5% lidocaine, 5% prilocaine) was distributed on the head. Animals were injected bilaterally with 1 µl of virus into the dorsal hippocampus

(coordinates from bregma: anterior/posterior −2.1 mm, medial/lateral ± 1.5 mm, and dorsal/ventral −1.8 mm) and 1 μl of virus in the ventral hippocampus (coordinates from bregma: anterior/posterior −3.3 mm, medial/lateral ± 2.75 mm, and dorsal/ventral −4.0 mm). Postoperative health checks were carried on over the 3 days after surgery.

## Tissue collection

Animals were sacrificed by intraperitoneal injection of pentobarbital (150 mg/kg). When in deep anesthesia, mice were perfused intracardially with ice-cold PBS pH 7.4, followed by perfusion with 4% paraformaldehyde in PBS pH 7.4. The brains were then isolated and postfixed for 2–3 hr (138-floxed; 138-sponge$^{PV}$ mice) or overnight (138-flox and 138-sponge$^{ub}$ mice) at 4°C. The fixed tissue was placed in sucrose solution (30% sucrose in PBS) for 24 hr and frozen in tissue mounting medium (OCT mounting media, VWR chemicals). The tissue was coronally sectioned at 50–60 μm thickness on a cryostat, immediately placed in ice-cold PBS, and subsequently conserved in cryoprotectant solution (15% glucose, 30% ethylene glycol, 5 mM $NaH_2PO_4*H_2O$, 20 mM $Na_2HPO_4*2H_2O$) at −20°C.

## Immunohistochemistry

Ex vivo (*cryosections*): for immunofluorescence, cryosections were washed in ice-cold PBS for 30 min, placed on microscope slides (Menzel-Gläser SUPERFROST PLUS, Thermo Fisher Scientific), and air-dried for 5–10 min. Afterward, permeabilization was performed by incubating sections in permeabilization solution (0.5% Triton X-100 in PBS) for 30 min at room temperature, followed by a blocking step in blocking buffer (0.5% Triton X-100, 300 mM NaCl, and 10% normal goat serum in PBS) with the addition of blocking Reagent (VC-MKB-2213, Adipogen Life Sciences, 1:10 dilution) for 1 hr at room temperature. Cryosections were washed in PBS two times for 10 min at room temperature and incubated with primary antibodies in blocking buffer overnight at 4°C. Subsequently, the sections were washed three times with blocking buffer, incubated with secondary antibodies and Hoechst 33342 in blocking buffer for 1 hr at room temperature, washed again three times in blocking buffer, washed in PBS, air-dried, and mounted with Aqua-Poly/Mount (POL18606, Polysciences).

Ex vivo (vibratome sections): for immunofluorescence, 300 μM sections were sliced on a vibratome in ice-cold sucrose-containing artificial cerebrospinal fluid, and then fixed in 4% PFA overnight. Free-floating sections were washed in ice-cold 1× PBS for 30 min, permeabilized by incubation in 0.5% PBX (0.5% Triton X-100 in PBS) for 30 min at room temperature, followed by a quenching step in 0.1 M Glycine to reduce background autofluorescence. Sections were blocked in blocking buffer (0.5% PBX, 10% normal goat serum in PBS) for 1 hr at room temperature, and then incubated in primary antibodies for 36 hr at 4°C. Sections were washed in 0.05% PBX two times for 10 min at room temperature, and then incubated with secondary antibodies in 0.05% PBX with 10% normal goat serum at room temperature for 3 hr.

Subsequently, the sections were washed three times with 0.05% PBX, incubated with Hoechst 33342 in 1× PBS for 10 min, and then washed again three times in PBS, mounted on SuperFrost + slides with Aqua-Poly/Mount (POL18606, Polysciences). A 40-μM Z-stack (with 0.25 μM z-step) was acquired on a Zeiss LSM 880 confocal microscope, with a Plan-Apochromatic 63×/1.4NA DIC M27 oil objective and FastAiryScan detector settings in the hippocampal CA1 region.

In vitro: cells were washed once with NBP, fixed for 20 min with 4%PFA and, after five washes with PBS (3× fast; 2× 5 min), permeabilized with GDB (20 mM sodium phosphate buffer (pH 7.4), 450 mM NaCl, 0.3% Triton X-100, 0.1% gelatin, and $ddH_2O$) for 20 min. Primary antibody was diluted in GDB and incubated for 1 hr at room temperature, followed by five washes with PBS (3× fast; 2× 5 min). Secondary antibody was diluted in GDB and incubated in the dark, at room temperature for 1 hr. The cells were then washed again with PBS (3× fast; 2× 5 min). In the second last wash, Hoechst 33342 was added to the PBS. The coverslips were mounted with Aqua-Poly/Mount (POL18606, Polysciences) on microscope slides (Menzel-Gläser SUPERFROST PLUS, Thermo Fisher Scientific).

## Statistics

Statistical analysis was performed on either GraphPad Prism 8.0 or RStudio. Plots were generated in R, mainly using the packages ggplot2, ggsci, ggrepel, and scales. For data sets with n>4, Box plots (Tukey style) were used. For data sets with n<5, average±s.d., including individual data points, is shown. Normal distribution of data was tested with the Shapiro-Wilk test. Based on the results,

parametric (e.g., Student's t-test) or nonparametric (e.g. Mann-Whitney and Kolmogorov-Smirnov) tests were used.

The detailed parameters (n, p value, test) for the statistical assessment of the data are provided in the figure legends.

## Acknowledgements

The authors are grateful to S Brown and M Müller for valuable comments on the manuscript, T Wüst and C Furler for excellent technical assistance, T Germade for help with bioinformatics, D Colameo for help with animal perfusions and microscopy data analysis scripts, A Loye for help with histology and T Demeter for help with cloning. The authors thank G Fishell, F Gage, and the Viral Vector Facility of the Neuroscience Center Zurich for sharing plasmids. This work was supported by grants from the DFG (SCHR 1136/4-2) and the ETH 24 18-2 (NeuroSno).

## Additional information

### Funding

| Funder | Grant reference number | Author |
|---|---|---|
| Deutsche Forschungsgemeinschaft | SCHR 1136/4-2 | Gerhard Schratt |
| Eidgenössische Technische Hochschule Zürich | 24 18-2 (NeuroSno) | Gerhard Schratt |

The funders had no role in study design, data collection and interpretation, or the decision to submit the work for publication.

### Author contributions

Reetu Daswani, Investigation, Methodology, Visualization, Writing - original draft, Writing - review and editing; Carlotta Gilardi, Prakruti Nanda, Kerstin Weiss, Silvia Bicker, Investigation; Michael Soutschek, Formal analysis, Software; Roberto Fiore, Investigation, Supervision; Christoph Dieterich, Formal analysis, Funding acquisition, Software, Supervision; Pierre-Luc Germain, Formal analysis, Software, Supervision; Jochen Winterer, Conceptualization, Investigation, Supervision, Visualization, Writing - review and editing; Gerhard Schratt, Conceptualization, Funding acquisition, Supervision, Visualization, Writing - original draft, Writing - review and editing

### Author ORCIDs

Michael Soutschek (iD) http://orcid.org/0000-0002-8472-8124
Silvia Bicker (iD) http://orcid.org/0000-0002-6276-5653
Christoph Dieterich (iD) http://orcid.org/0000-0001-9468-6311
Pierre-Luc Germain (iD) http://orcid.org/0000-0003-3418-4218
Jochen Winterer (iD) http://orcid.org/0000-0002-6800-6594
Gerhard Schratt (iD) http://orcid.org/0000-0001-7527-2025

### Ethics

All procedures were conducted in strict accordance with the National Institutes of Health Guidelines for the Care and Use of Laboratory Animals and the relevant local or national rules and regulations of Switzerland and were subject to prior authorization by the local cantonal authorities (ZH017/2018, ZH196/17).

### Decision letter and Author response

Decision letter https://doi.org/10.7554/eLife.74056.sa1
Author response https://doi.org/10.7554/eLife.74056.sa2

## Additional files

### Supplementary files
• Transparent reporting form

### Data availability
RNA-seq data has been deposited to GEO (accession no. GSE173982).

The following dataset was generated:

| Author(s) | Year | Dataset title | Dataset URL | Database and Identifier |
|-----------|------|---------------|-------------|-------------------------|
| Daswani R, Dieterich C, Schratt G | 2021 | miR-138 controls interneuron function and short-term memory | https://www.ncbi.nlm.nih.gov/geo/query/acc.cgi?acc=GSE173982 | NCBI Gene Expression Omnibus, GSE173982 |

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
