## [Editor Report]

The authors provide evidence for a role of the micro RNA – mi-138 – in synaptic inhibition and behavior in mice. They designed a novel transgenic model in which a miR-138 sponge is conditionally expressed to sequester endogenous miR-138 in a cell-type specific manner. Using this tool they show that sequestration of miR-138 in a major, parvalbumin-expressing inhibitory cell type results in enhanced inhibitory transmission, and deficits in memory.

---

## [Decision Letter]

[Editors' note: this paper was reviewed by Review Commons.]

---

## [Author Response]

Reviewer #1 (Evidence, reproducibility and clarity (Required)):Daswani et al. study the role of miR-138-5p, a microRNA the authors previously found to regulated excitatory synapse function, in mutant mice. To this end they overexpress a miR-138 sponge construct first ubiquitously and find that short term memory is impaired. Gene-expression analysis points to a specific role in inhibitory neurons, which the authors confirm by repeating key experiments in mutant mice in which the miR-138 sponge is expressed in PV-inhibitory neurons. This is a very interesting study that pioneers the analysis of microRNAs in a cell-type specific manner in the CNS. The following point represent helpful suggestion the authors could address before submission.- A question that may come up is related to the mutant mice the controls. One could argue that the better control would the transgenic mice expressing the "control sponge". At present the control group are mice that carry the construct but its not expressed, as far as I understood. The authors may want to address this issue upfront and discuss somewhere in the manuscript that their control animals are suitable.

We agree with this reviewer that an independent line expressing a “control sponge” construct might represent a “gold standard” control. However, we decided to take the current approach for two reasons: (1) considering the RRR principles of animal research, including an independent mouse line would have doubled the number of animals needed for experiments. (2) although a control sponge line addresses potential specificity issues of expressing a sponge transcript, it also has its own limitations since behavioural experiments cannot be done with littermates, which could also introduce variability. Nevertheless, we have rigorously validated the specificity of the miR-138 sponge construct by comparing it to a highly similar miR-138 control sponge construct in several assays: (i) miR-138 perfect binding site luciferase sensor assay (suppl. Figure 1A); (ii) dendritic spine size assay (suppl. Figure 1B); (iii) mIPSC electrophysiological recordings from hippocampal slices of intracerebrally injected mice (new Figure 4d, e); (iv) short-term memory assay in intracerebrally injected mice (new Figure 4b, c); (v) Erbb4 FISH in primary rat hippocampal neurons infected with Dlx-138-sponge (new Figure 3f). As suggested by this reviewer, we have discussed our approach in more detail in the revised manuscript (p.10-11, lines 307-316).

- Moreover, as far as I understand it, the sponge construct will be expressed from early developmental stages in all cells of the body in the initial experiment. While the authors later study specifically PV neurons, this may also represent early neurodevelopmental effects. It might be helpful for the reader to explain the approach and rational in more detail. want the specifically focus on the role of miR-138 in neurodevelopmental diseases (schizophrenia is generally mentioned in the introduction) in which case the approach would make sense. However, I am not aware that miR-138 has been genetically linked yet to neurodevelopmental disease but I might have missed this. This could clarified specifically in the introduction.

We agree with this reviewer that the phenotypes we observe might have an early developmental origin, at least in the ubiquitous sponge mouse line. However, as pointed out by reviewer 2, expression of PV doesn’t kick in before the second postnatal week, arguing for a postnatal role of miR-138-5p in the PV-sponge model. Moreover, stereotactic injections of Dlx-138-sponge into the hippocampus (new Figure 4) were performed in young adult mice, providing further support that perturbing miR-138-5p at adult stages impacts physiology and behaviour. Our findings are thus in line with a link to a disease like schizophrenia whose onset is either during adolescence (early-onset) or in adulthood (late-onset). In fact, miR-138 has been linked to cognitive functions, and a rare variant in the miR-138-2 gene has been associated with schizophrenia (Watanabe et al., Psychiatry Res. 2014). We have now better clarified these links in the introduction and discussion of our revised manuscript. In addition, we have provided additional data showing a preferred deregulation of SCZ-associated genes in the miR-138 sponge model (new suppl. Figure 4d; see also below).

- Figure 1B. To convincingly demonstrate the functionality of the construct in vivo it may help to assay the regulation of some confirmed targets at the mRNA/Protein level or point already here to the exp. later shown in Figure 2.

We agree with this reviewer, that showing target gene regulation is imperical to confirm the functionality of the miR-138 loss-of-function approach. However, we want to stress that a number of experiments in this context were already included in our original manuscript. (1) Expression of the sponge in mice in vivo leads to a preferential upregulation of predicted miR-138 targets compared to non-targets (Figure 2b), which provides strong evidence that expression of the construct leads to a specific loss-of-function of miR-138. (2) Validation by qPCR confirmed upregulation of PV-enriched genes at the RNA level, e.g. Rims3 and Erbb4. (3) In the revised manuscript, we further included protein data for Erbb4 upregulation in the hippocampus of miR-138-sponge^Ub^ mice using Western blot (new Figure 3c). Unfortunately, we were unable to obtain conclusive Western blot results for Rims3 using several commercially available antibodies.

- Supplemental Figure 1.: n=1 for pGL3 CTR may not be sufficient for statistical analysis. Panel (A) Please explain this assay in greater detail. What is measured here?

In fact, this assay has been repeated 3 times. The lack of error bar in the control conditions is due to normalization. Statistical test could still be performed with a 1-sample t-test, which revealed a significant difference. The experiment measures the endogenous activity of miR-138 in hippocampal neurons (based on the extent of cleavage of the luciferase reporter).

- Supplemental Fig1d: No scale bar. Why show a small section of the cerebellum when later hippocampus is analyzed? I suggest to show a selection of brain regions with high quality images.

We agree and have now included a lacZ staining from the hippocampus in the revised manuscript (new suppl. Figure 1d).

- Figure 1. The authors may want to consider that no effect in the contextual fear conditioning may not be sufficient to claim that specifically short-term memory is affected in mutant mice. A more thorough behavioral characterization may be necessary or it should be discussed why this is not necessary in the context of the current manuscript.

In the revised manuscript, we have now further included CFC data from the PV-sponge mouse line (new Suppl. Figure 5d), which also didn’t reveal a significant genotype-dependent difference. That said, we fully agree that results from only one behavioural paradigm (CFC) are not sufficient to make general statements about the specific types of memories affected by miR-138 loss-of-function. However, since our laboratory is currently not set up to perform an extensive behavioural characterization of memory function using different tests, we consider such experiments beyond the scope of this revision. We therefore decided to tone down the statements in the manuscript related to short-term memory specificity accordingly.

- Figure 2: please explain in more detail (in the methods) the statistics behind the diff. expression analysis.

A more detailed statistical description has been included in the methods section of the revised manuscript (p. 18, l. 1074-1079).

- Figure 2d. To further strengthen the idea that DEG are enriched in inhibitory neurons, corresponding viol plots across cell types may help. The authors could plot for example selected genes are the eigen-vector expression of DEGs.

We agree with this reviewer that an alternative presentation of the DEG data could be helpful. We have therefore added another panel (new suppl. Figure 2c) illustrating the distribution of expression of the sponge-upregulated-DEGs in the different cell types. In our view, this presentation is more easily interpretable compared to viol plots.

- The selection of Erbb4 and Rims3 for further analysis could be explained better.

Based on the available data, we have now decided to prioritize on Erbb4 regulation in the revised manuscript (Rims3 data was moved to the supplement). We have further provided a more detailed rationale for studying Erbb4 in the context of inhibitory synaptic transmission and short-term memory.

- Figure 3B.Maybe show representative images.

Representative pictures have now been included in the revised manuscript (new Figure 5b, left panel).

- How about FC and other long-term memory assays in 138-spongePV´ mice?

We have now additionally performed CFC in 138-sponge-PV mice, and found no significant difference in memory performance between 138-sponge and control mice (new suppl. Figure 5d). As outlined above, additional memory assays are beyond the scope of this study.

- Please address the finding that behavior phenotypes are similar in 138-spongeub and 138-spongePV´ mice, while e-physiology differs (mini ipsc).

We assume that this is a misunderstanding, since we did not measure mIPSCs in pyramidal neurons of 138-sponge^Ub^ mice (in contrast to mEPSC, which surprisingly were not altered compared to control mice.)

- I guess an obvious question could be to show the regulation of miR-138 target genes in PV neurons and /or study the transcriptome in PV-neurons as a result of sponge expression.

We agree that assessing the transcriptome specifically of PV neurons would be a logical next step. However, this is technically rather challenging, either involving FACS sorting of PV-positive neurons or single-cell RNAseq approaches, both of which are currently not established in the laboratory. We therefore feel that such experiments, also considering the limited new insight that would be expected from them, are beyond the scope of this revision.

- The discussion is comparatively short. The specific reference to miR-138 as a target for schizophrenia and ASD provokes the question of more detailed analysis, e.g behavioral phenotypes for ASD and schizophrenia in mutant mice or comparing the DEG to genes de-regulated in other mouse models for these diseases.

We agree that making more functional connections to ASD/schizophrenia would be exciting. However, this would involve a battery of additional mouse behavioural experiments, which is in our view clearly beyond the scope of this revision. Concerning the second point, we have performed a comparison between DEGs deregulated in the miR138 sponge model and a previous study about schizophrenia (Gandal et al., Science 2018). Excitingly, we found that the genes upregulated in the 138-sponge (at FDR<0.05) are significantly enriched for genes found upregulated in the brain of SCZ patients (p~0.00884; new suppl. Figure 4d), which provides additional correlative evidence for an involvement of miR-138 in SCZ. Moreover, we have elaborated more on the role of miR-138 in cognition and SCZ in the discussion (p. 11-12, l. 339-351).

- Given that the literature suggests of role for miR-138 also excitatory neurons, the phenotypes should be discussed in more detail. Why are mainly genes in inhibitory neurons affected in 138-spongeub mice?

Based on our new results from a cell-type specific miR-138 sensor assay (new suppl. Figure 4c) we speculate that the reason for the preferential regulation in inhibitory neurons is the more efficient miR-138 inhibition in this cell type. This is now discussed in more detail (p. 10, l. 299-306). Please see also our comments to the other reviewers regarding this topic.

Reviewer #1 (Significance (Required)): Generally this study addresses the cell-type specific role of microRNAs in the CNS, which is important to still innovative.The finding that miR-138 controls specific functions in inhibitory neurons and that this is linkned to behavioral alterations is novelThe data will be mainly intersting for the neuroscientific communityReviewer #2 (Evidence, reproducibility and clarity (Required)):Understanding the role of miRNAs in different neuronal cell types remains a paramount goal in neuroscience. Using a conditional miR-138 sponge transgenic line, Daswani and colleagues demonstrate that miR-138 in parvalbumin+ (PV) GABAergic neurons is necessary for short-term memory. Further, they identify a possible cellular mechanism underlying this behavioral deficit: CA1 pyramidal neurons receive more inhibition from neighboring PVs. This work is timely, impactful, and an important contribution to the field. However, there are important aspects that needs to be addressed before publication:Major:It is surprising that the miR-138 sponge has no effect in pyramidal neurons. The authors' laboratory has published beautiful work showing that miR-138 has important roles for synaptogenesis in this cell type. The authors hypothesize that the expression levels of the miR-138 sponge might not be high enough for pyramidal neurons (where miR-138 is highly expressed) and instead it is sufficient for PVs. However, sensor data in Figure 1b and spines data in Sup Figure 1c are from pyramidal neurons. For the miR-138 sponge to be credible, the authors must demonstrate that, as they claim, the sponge is effectively working in PVs and only partially in pyramidal neurons. This could be achieved by showing that miR-138 is expressed at different levels in the two cell types: (i) analysis of miR-138 expression levels from He et al. (2012) in different cell types; (ii) FACS of PVs and pyramidal neurons and assessing miR-138 levels; (iii) quantifying miRNA FISH in different cell types. In addition, the authors should demonstrate that the miR-138 has different levels of efficacy in the two cell types, i.e. analyzing sensor data in PNs vs PVs (from cortical cultures or in vivo). If the results from this further analysis don't support the original hypothesis, the authors must find a plausible reason and validate it.

We fully agree with this reviewer that the absence of phenotype in pyramidal neurons is surprising and that we do not have a full explanation for this yet. To further test our hypothesis of an incomplete miR-138 inhibition in excitatory pyramidal neurons of the miR-138 sponge line, we have now performed a series of new experiments, largely following the suggestions made by this reviewer: (i) we monitored miR-138 expression from a published small RNA-seq study (He et al., 2012) (new suppl. Figure 3b). (Ii) we quantified miR-138 signal in already existing smFISH datasets from hippocampal neurons containing PNs and PVs (new suppl. Figure 3a). (Iii) we quantified mir-138 sensor activity in PNs and PVs in hippocampal slices stained for PN and PV markers (new suppl. Figure 4c). Altogether, this analysis suggests that while miR-138 is expressed to a similar level in PNs and PVs, its activity is markedly higher in PVs and more robustly inhibited in PVs by expression of a miR-138 sponge. Although the mechanistical underpinnings are unknown, these observations might explain why we preferentially observed miR-138-5p dependent regulation of interneuron-enriched genes in miR-138^Ub^ sponge mice.

Minor:The sensor data are somewhat confusing. It would benefit to add schemes of the sensor and sponge constructs and to represent the data in Figure 1b with a bar graph. To further support the claims of incomplete effects of the miR-138 sponge, the authors should have a mutated MRE sensor (maximum de-repression) and express the effect of the sponge in different cell types as percentage of maximum de-repression. Please also add text explaining the sensor experiments in more detail so that the non-specialist can follow.

As suggested by this reviewer, we have now added a scheme of the sensor and presented the data in Figure 1b as a bar graph. In addition, we have performed the sensor analysis separately for PNs and PVs (new suppl. Figure 4c). This analysis clearly shows that the degree of inhibition exerted by the 138-sponge is more pronounced in PV+ interneurons compared to PV- neurons, thereby supporting our model that transgenic expression of the miR-138 sponge preferentially de-represses miR-138-5p target genes in interneurons.

The choice of PV-Cre is interesting. PV-Cre does not kick in until the second postnatal week, it is notoriously not a very efficient Cre line, and although it correlates well in the cortex with basket cells, it turns on also in cerebellar purkinje cells and in the peripheral nervous system. Some of these limitations should be discussed, considering that behavior is used as readout.

The point about ectopic expression of PV-Cre in PV- cells in cerebellar purkinje cells and the peripheral nervous system is well taken. We therefore decided to perform a complementary approach where we delivered 138-sponge and control-sponge specifically to hippocampal interneurons by stereotactic injection of a rAAV-Dlx5/6 construct (new Figure 4). Intriguingly, the miR-138-sponge associated behavioural and electrophysiological phenotypes were fully recapitulated by this approach. Together with our detailed electrophysiological characterization of hippocampal PV+ interneurons in paired recordings (Figure 6), we are confident that hippocampal PV dysfunction contributes in an important way to the observed phenotypes.

The authors should remove emphasis from E/I. The physiology experiments performed don't assess E/I but E and I in isolation. It is the opinion of this reviewer that the work is impactful because it reveals novel miRNA mechanisms in GABAergic cells, no need to claiming anything else.

We agree that our claims about E/I are currently overstated, and have now toned down the respective statements in the revised manuscript (e.g. by removing emphasis from E/I in the introduction).

Sup Figure 1c. Should the y axes say something like Spines Density?

We have added the y-axes label as suggested.

Sup Figure 1d. Why did the authors choose the cerebellum to show CMV-Cre efficacy? The rest of the paper focuses on the hippocampus.

We have now provided a new lacZ staining from the hippocampus to illustrate efficient Cre-mediated recombination in this region (new suppl. Figure 1d).

In the introduction, the Tan et al. paper from the Schaefer lab should be mentioned, considering that miR-128 was knocked out in GABAergic cells (MSNs).

We thank the reviewer for this reminder and have now referenced this paper in the introduction.

Reviewer #2 (Significance (Required)): Identifying cell type-specific roles of miRNAs has eluded the field, specifically in the case of rare cell types, i.e GABAergic interneurons. The paper from Daswani and colleagues is an important step forward in this direction. Somewhat serendipitously, the authors have found that miR-138 targets preferentially genes enriched in GABAergic interneurons. Indeed, PV-specific knockout of miR-138 recapitulates behavioral deficits induced by global miR-138 knockout. The only similar contributions are well referenced in the introduction: (i) Tuncdemir et al. (Fishell lab) performed MGE-specific Dicer knockout and found defects in GABAergic interneuron survival, migration, and lamination; (ii) Qiu et al. (Miao He lab) showed that Dicer ko in VIP cells induces long-term dysfunction in GABAergic interneuron computations and survival.This manuscript should be of interest to both the miRNA field and the synaptic plasticity, learning and memory fields.I am a neurobiologist that studies the roles of miRNAs in instructing network formation.Reviewer #3 (Evidence, reproducibility and clarity (Required)):Daswani et al. report an interesting role for miR-138-5p in the control of short-term memory and inhibitory synaptic transmission. They notably built on an elegant transgenic model in which a miR-138 sponge is conditionally expressed using the Cre-Lox system and allows the sequestering of endogenous miR-138 in a cell-type specific manner. Using this model, the authors provide evidence that miR-138-5p expressed in parvalbumin-expressing neurons controls short-term memory as well as inhibitory synaptic input onto CA1 pyramidal neurons. Using RNAseq and qrtPCR, they identify presynaptic genes whose expression is regulated by miR-138 and possibly accounting for inhibitory synaptic alterations. Overall, the experiments are well conducted. The data are interesting, solid and logically presented and the methods are sufficiently described. However, there are some concerns that should be considered prior publication:Major concerns:(1) In the abstract, the authors' claim that "miR-138-5p regulates the expression of presynaptic genes in hippocampal parvalbumin-expressing inhibitory interneurons to control short-term memory" is overstated. Indeed, while the authors provide compelling evidence that miR-138 sponge controls short-term memory, the causal evidence to demonstrate that this is going through the regulation of presynaptic genes expression is missing. Moreover, because they are using a transgenic model in which the 138-sponge is constitutively expressed in any cell types and brain areas, the short-term memory deficits found by the authors are not necessarily resulting from hippocampal synaptic alterations. Therefore, the authors should either tone down this claim or perform additional experiments to support their model (see points 3 and 4).

We largely agree with these statements and specifically address them under points 3 and 4.

(2) While the authors provide convincing data demonstrating the efficiency of the miR-138 sponge both in cultured neurons and in vivo using sensor constructs and luciferase assays, the only evidence that the 138-sponge specifically inhibits miR-138 is by using a control sponge in cultured neurons and by measuring luciferase activity from sensors or spine size. To better support the specificity of the 138-sponge, it is important to demonstrate the absence of effect of the control sponge on inhibitory synaptic transmission, at least in cultured neurons.

As suggested by this reviewer, we have now performed additional experiments to support the specificity of the 138-sponge in interneurons. Therefore, we generated rAAV expressing 138-sponge selectively in interneurons due to the presence of a Dlx5/6 promoter (Dlx-138-sponge). (i) expression of Dlx-138-sponge, but not Dlx-control-sponge, elevated Erbb4 expression in rat hippocampal interneurons in culture as assessed by smFISH (new Figure 3f); (ii) stereotactic injection into the adult mouse hippocampus of Dlx-138-sponge, but not Dlx-control-sponge, impairs short-term memory (new Figure 4b, c) and inhibitory synaptic transmission in pyramidal neurons (new Figure 4d, e).

Moreover, it would be reassuring to see that not all miRNAs are regulated by miR-138 (for instance by using sensors for distinct miRNAs).

A priori, one cannot rule out that miR-138 inactivation indirectly regulates the expression and/or activity of other miRNAs, for example by regulating targets which control miRNA biogenesis. Therefore, the proposed sensor experiments for selected miRNA candidates are in our view not suitable to unequivocally judge the specificity of the miR-138 sponge approach. Moreover, such experiments would have required the stereotactic injection of at least two additional cohorts of mice, which was not covered by our currently approved animal experimentation license. Finally, only a low number of miRNAs could be screened with such an in vivo approach, making general conclusions difficult. By weighing all these arguments, we decided not to perform sensor experiments for additional miRNAs and instead added further experiments which support the specificity of the 138-sponge approach (see above).

(3) The authors propose that miR-138 expression in the hippocampus regulates short-term memory. Although there is evidence from previous studies that hippocampal PV+ interneurons are involved in short-term memory, the data do not directly address this point. Specific involvement of miR-138 in hippocampal PV+ interneurons could be easily tested through stereotaxic injections of rAAVs carrying PV-Cre in 138-floxed mice in order to drive the expression of miR-138 sponge selectively in CA1 hippocampal interneurons.

We agree that the data presented in the original manuscript didn’t provide direct evidence for a role of the hippocampus in miR-138-dependent regulation of short-term memory. Therefore, we now inactivated miR-138-5p specifically in hippocampal interneurons by stereotactic injection of a Dlx-138-sponge rAAV construct (new Figure 4a, see above). This led to a similar impairment in both short-term memory (new. Figure 4b, c) and inhibitory synaptic transmission (new Figure 4d, e) as observed in 138-sponge^Ub^ and 138-sponge^PV^ mice. We did not pursue the PV-Cre injections into 138-floxed mice, since viral expression of PV-Cre (as opposed to transgenic expression in PV-Cre mice) apparently is not very specific and therefore not frequently used (personal communication).

(4) Although not required for the paper as it is, experiments addressing the causal link between (1) the miR-138-dependent regulation of ErbB4 and/or RimS3 in CA1 interneurons and (2) the role of miR-138 in inhibitory synaptic transmission and/or short-term memory would considerably strengthen the model proposed. To address that, the authors could alter the expression/function of the candidate genes in their transgenic model. Alternatively, strategies could be used to selectively protect putative target from the action of miR-138-5p (e.g., using LNAs or 3'UTR mutants).

Addressing the molecular mechanism underlying impaired short-term memory and inhibitory transmission is clearly the next step in this project, but beyond the scope of the current manuscript, as also pointed out by this reviewer.

(5) Description of statistics require clarifications: - n should be clearly defined in all figure legends (cells, cultures, animals, experiments?).- The authors should indicate whether and how they tested for normal distribution of their data.—figure supplement Figure 1a-c and Suppl. Figure 3a : a non-parametric test should be used to compare the different conditions, considering the too low n number which does not allow to conclude on a normal distribution.

We have now addressed these statistical issues in the revised manuscript. (i) we have now clearly defined n in all figure legends. (ii) normal distribution of the data was tested with the Shapiro-Wilk test in GraphPad (stated in the Methods section). When normal distribution was not observed, non-parametric tests were used (e.g. Mann-Whitney, Kolmogorov-Smirnov). (iii) we agree with this reviewer that the n for the indicated experiments is too low to conclude on a normal distribution with a common test. However, visual inspection of the data suggested normal distribution for luciferase assays presented in Suppl. Figure 1a-c, considering our long-term experience with these assays. We therefore did not see the necessity to use non-parametric tests in this case. Concerning the PV+ cell density count (previous Suppl. Figure 3a, new suppl. Figure 6e), we now applied the Mann-Whitney test.

Minor issues: (1) The fact that excitatory synapses onto pyramidal neurons are unaffected in 138-sponge mice is puzzling when considering the previous study from the same authors (Siegel et al., 2009) as well as the results obtained in cultured neurons to validate the sponge (spine size measurements, Suppl. Figure 1). While the authors propose that 'the absence of changes in excitatory synaptic transmission in the hippocampus of 138-sponge-ub mice might be due to ineffective silencing of the highly abundant miR-138-5p in pyramidal neurons' they could actually refine their analysis to give support this hypothesis:- By quantifying the signal from single-molecule FISH for miR-138 in Camk2a+ neurons versus Erbb4+ neurons (Figure 2e). In the images shown, it is not evident that excitatory neurons express more miR-138 in comparison to inhibitory ones.- By quantifying the degree of inhibition of miR-138 in pyramidal neurons versus interneurons using the dual sensor system (GFP/mcherry signal). In Figure 1b, the authors do not discriminate between cell types. A lack of increase of the GFP signal in pyramidal cells may indicate an inability of those cells to express the sponge.- By checking whether APT1, a known miR-138 target that they previously identified in excitatory neurons (Siegel et al., 2009) is also a regulated gene in 138-sponge mice. If APT1 expression remains unchanged, this could give support to the fact that the sponge is indeed ineffective in excitatory neurons.

We agree with the comments made by this reviewer, which are also in agreement with those from reviewer 2 (see also our comments there). Concerning the last point of reviewer 3, we now checked the expression APT1 (new nomenclature LYPLA1), as well as two other validated miR-138 targets from excitatory neurons (SIRT1, Reln) in our RNA-seq data. None of these targets was differentially expressed (new suppl. Figure 2e). Together, this data supports our hypothesis that miR-138 is not effectively inhibited by the sponge transgene in PNs.

(2) The analysis of VGAT signal intensity in individual PV+ boutons may provide useful information about the content in vesicles per bouton. One could expect a higher VGAT intensity/bouton in 138-sponge mice, correlated with an increase number of release sites per bouton (as proposed by the authors) or a similar trend as for increased success rate or decreased PPR.

We agree that the analysis of VGAT signal intensity might provide additional information regarding the observed inhibitory synapse phenotype and therefore performed the respective analysis (new Figure 6b, right panel). However, this didn’t reveal any significant differences between miR-138-sponge^PV^ and control mice, suggesting that the gross morphology of inhibitory release sites is unchanged. Further ultrastructural investigation of synaptic release sites will require super-resolution microscopy, which is currently not established in the laboratory and therefore beyond the scope of this revision.

(3) For the following data, titles of the graph axis are not accurate and the authors should indicate more precisely what are the exact parameters measured: Figure 1b, Suppl. Figure 1a, Suppl. Figure 1c, Suppl. Figure 3b.

We have fixed these issues in the revised manuscript.

(4) Figure 3j: "Coefficient of variance" should be corrected for "coefficient of variation".

We have changed this expression accordingly.

(5) Results – L70: the use of the miR-138 sponge is critical for the key findings of the study. It is therefore important to detail within the text how the specificity and the efficiency of the sponge were tested.

Please refer to our comments for reviewer 1 and 2 regarding miR-138 sponge validation in vitro and in vivo. In addition, we have provided more discussion of the sponge strategy in the text (p. 10-11, l. 307-316).

(6) Results – L75: the authors should introduce briefly how the dual sensor works and why an increase of the GFP signals indicate efficient sequestering of endogenous miR-138.

We have now explained the sensor strategy in more detail in the Results section (p.4, l.98-106) and in addition provided a schematic (new suppl. Figure 1e)

(7) Supplementary figure 1d: to illustrate the 'penetrant expression' of the sponge, the authors should provide an image representative of the hippocampus not the cerebellum.

We have now provided a representative picture of the hippocampus (new suppl. Figure 1d).

(8) Suppl. Figure 1g and Suppl. Figure 3B: the use of a single two-way ANOVA test with a single P value is not clear to me as two different parameters are measured (distance and time).

We apologize for the confusion. The two parameters have now been plotted as individual panels (new suppl. Figure 1g, h; suppl. Figure 5a, b) and separately analyzed with heteroscedastic t-tests.

(9) Suppl. Figure 1i,j and suppl. Figure 3d: the authors should mention whether a statistical test was used to compare the two conditions. n values (cells) should be indicated.

The respective figures correspond to bar graphs in the main manuscript (Figure 1, 6), for which a statistical test was provided. However, we now also provided a statistical assessment of the cumulative graphs (using KS-test) in the revised manuscript.

Reviewer #3 (Significance (Required)): This study provides important findings that will be of interest for a large audience of neuroscientists interested in synaptic physiology and pathologies:(1) The authors developed an elegant transgenic model allowing to address the role of miR-138 in a cell specific manner and which allows to correlate synaptic alterations to the behavior.(2) While the majority of studies have investigated the role of miRNAs at excitatory synapses, this study provides important insight about the role of one identified miRNA, miR-138, at inhibitory synapses. Interestingly, because miR-138 was also shown to regulate excitatory synapses, the data suggest a central role of miR-138 in regulating E-I balance in hippocampal circuits.(3) The study further shows that alterations in inhibitory synaptic transmission resulting from miR-138 inhibition are correlated with deficits in working memory. Because miR-138 as well as its targets are associated with cognitive deficits and neuropsychiatric disorders such as autism and schizophrenia (Kumar et al., 2010; Nicodemus et al., 2006), this study thus suggests a critical role of miR-138 in disorders with altered E/I balance.I have expertise in the role of microRNAs in synaptic development and plasticity. While I have little expertise regarding behavioral investigations in general, I feel that the behavioral data shown in the present paper are easily accessible and straightforward.